

# Forecasting experiments of a dynamical-statistical model of the sea surface temperature anomaly field based on the improved self-memorization principle

Mei Hong[1,2], Xi Chen[1], Ren Zhang[1,2], Dong Wang[3], Shuanghe Shen[2], and Vijay P. Singh[4]

[1]*Institute of Meteorology and Oceanography, National University of Defense Technology, Nanjing 211101, China*

[2]*Collaborative Innovation Center on Forecast and Evaluation of Meteorological Disaster, Nanjing University of*

*Information Science &Technology, Nanjing 210044, China*

[3]*Key Laboratory of Surficial Geochemistry, Ministry of Education; Department of Hydrosciences, School of Earth*

*Sciences and Engineering, Collaborative Innovation Center of South China Sea Studies, State Key Laboratory of*

*Pollution Control and Resource Reuse, Nanjing University, Nanjing 210093, China*

[4]*Department of Biological and Agricultural Engineering, Zachry Department of Civil Engineering, Texas A & M*

*University, College Station, TX 77843, USA*

Corresponding authors address:

1.  Xi Chen, Research Centre of Ocean Environment Numerical Simulation, Institute
    of Meteorology and Oceanography, National University of Defense Technology,
    Nanjing 211101, China

E-mail: chenxigfkd@163.com

2.  Ren Zhang, Research Centre of Ocean Environment Numerical Simulation,
    Institute of Meteorology and Oceanography, National University of Defense
    Technology, Nanjing 211101, China

E-mail: 254247175@qq.com



**Abstract:** With the objective of tackling the problem of inaccurate long-term El Niño
Southern Oscillation (ENSO) forecasts, this paper develops a new
dynamical-statistical forecast model of sea surface temperature anomaly (SSTA) field.
To avoid single initial prediction values, a self-memorization principle is introduced
to improve the dynamic reconstruction model, thus making the model more
appropriate for describing such chaotic systems as ENSO events. The improved
dynamical-statistical model of the SSTA field is used to predict SSTA in the
equatorial eastern Pacific and during El Niño and La Niña events. The long-term
step-by-step forecast results and cross-validated retroactive hindcast results of time
series $T_1$ and $T_2$ are found to be satisfactory, with a correlation coefficient of
approximately 0.80 and a mean absolute percentage error of less than 15%. The
corresponding forecast SSTA field is accurate in that not only is the forecast shape
similar to the actual field, but the contour lines are essentially the same. This model
can also be used to forecast the ENSO index. The correlation coefficient is 0.8062,
and the MAPE value of 19.55% is small. The difference between forecast results in
summer and those in winter is not high, indicating that the improved model can
overcome the spring predictability barrier to some extent. Compared with six mature
models published previously, the present model has an advantage in prediction
precision and length, and is a novel exploration of the ENSO forecast method.
**Keywords:** Dynamical-statistical forecast model; self-memorization principle; sea
surface temperature field; long-term forecast of ENSO

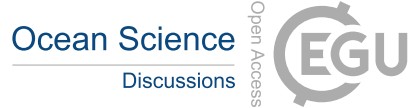

## 1. Introduction

The El Niño Southern Oscillation (ENSO), the well-known coupled atmosphere
–ocean phenomenon, was firstly proposed by Bjerknes (1969). The ENSO
phenomenon can influences regional and global climates, so the prediction of ENSO
has received considerable public interest (Rasmusson and Carpenter, 1982; Glantz et
al., 1991).
Over the past two to three decades, one might reasonably expect the ability to
predict warm and cold episodes of ENSO at short and intermediate lead times to have
gradually improved (Barnston et al., 2012). Many countries have been focusing on
ENSO forecasts since the 1990s, and the ENSO forecast has become one of the
important research topics in the International Climate Change and Predictability
Research plan. The U.S. International Research Institute for Climate and Society, the
U.S. Climate Prediction Centre, Japan Meteorological Agency, and European Centre
for Medium-Range Weather Forecasting have developed different coupled
atmosphere–ocean models to forecast ENSO (Saha et al., 2006; Molteni et al., 2007) .
The forecast models can generally be divided into two types (Palmer et al., 2004).
The first type is typified by a dynamic model, which mathematically expresses
physical laws that govern how the ocean and the atmosphere interact. The second type
is typified by a statistical model, which requires large a amount of historical data and
analyses the data to do forecasting (Chen et al., 1995; Moore et al., 2006).
Over the past three decades, ENSO predictions have made remarkable progress,
reaching a stage where reasonable statistical and numerical forecasts (Jin et al.,





2008)can be made 6–12 months in advance (Wang et al., 2009). . However, there are
three problems remaining to be resolved (Zhang et al., 2003a): (1) The current ENSO
predictions are mainly limited to the short term, such as annual and seasonal
predictions; (2) Although the representation of ENSO in coupled models has
advanced considerably during the last decade, several aspects of the simulated
climatology and ENSO are not well reproduced by the current generation of coupled
models. The systematic errors in SST are often very large in the equatorial Pacific,
and model representations of ENSO variability are often weak and/or incorrectly
located (Neelinet al. 1992; Mechoso et al. 1995; Delecluse et al. 1998; Davey et al.
2002). (3) Coupled models of ENSO predictions initialized from observed initial
states tend to adjust towards their own climatological mean and variability, leading to
forecast errors. The errors associated with such adjustments tend to be more
pronounced during boreal spring, which is often called the ''spring predictability
barrier'' (Webster et al., 1999). More efficient models are therefore desired (Belkin
and Niyogi, 2003; Weinberger and Saul, 2006). Therefore, the idea of combining
dynamical and statistical methods to improve weather and climate prediction has been
developed in many studies (Chou, 1974; Huang et al., 1993;Yu et al., 2014a; Yu et
al.,2014b). By introducing genetic algorithms (GAs), Zhang et al. (2006) inverted and
reconstructed a new dynamical-statistical forecast model of the tropical Pacific sea
surface temperature (SST) field using historic statistical data (Zhang et al., 2008).
However, there is one flaw in the forecast model: the time-delayed SST field. This is
because ENSO is a complicated system with many influencing factors. To overcome



information insufficiency in the forecast model, Hong et al. (2014) selected the
tropical Pacific SST, SSW and SLP fields as three modelling factors and utilized the
GA to optimize model parameters.

However, the above dynamical prediction equations which were ,proposed by

Hong et al.(2014), greatly depend on a single initial value, creating long-term
forecasts over 8 months that diverged significantly. These unsatisfactory results
indicate that this model needs to be improved. Cao (1993) first proposed the
self-memorization principle, which transforms the dynamical equations with the
self-memorization equations, wherein the observation data can determine the memory
coefficients. This method has been widely used in forecast problems in environmental,
hydrological and meteorological fields (Feng et al., 2001; Gu, 1998; Chen et al.,
2009). The method can avoid the question of initial conditions for the differential
equations, so it can be introduced here to improve the proposed dynamical forecast
model.

Therefore, an improved dynamical-statistical forecast model of the SST field

and its impact factors with a self-memorization function was developed. The
improved model can absorb the information from past observations.

This paper is organized as follows: Research data and forecast factors are

introduced in section 2. In Section 3 the reconstruction of the dynamical model of
SSTA field is described. To improve the reconstruction model, the self-memorization
principle is introduced in Section 4. Model forecast experiments are described in
Section 5, and conclusions are given in Section 6.



## 2. Research data and forecast factors

### 2.1 Data

The monthly average SST data from January 1951 to January 2010, 720 months in total, were obtained from the UK Met Office Hadley Centre for the region (30 °S-30 °N; 120 °E -90 °W). The sea areas provide important information on ocean-atmosphere coupling in the East and West Pacific Ocean and the El Niño and La Niña events. The reanalysis data and zonal winds were obtained from the National Center for Environmental Forecast (NECP) of America and the National Center for Atmospheric Research (NCAR) (Kalnay et al., 1996). The SOI data were obtained from the Climate Prediction Center (CPC). The time series of all data were from Jan. 1951 to Jan. 2010.

### 2.2 EOF deconstruction

The sea surface temperature anomaly (SSTA) field can be calculated from the SST field and can be deconstructed into time (coefficients)-space (structure) using the EOF method. Detailed information on the EOF method can be seen in the related references (Dommenget & Latif, 2002).

An empirical orthogonal function (EOF) analysis of smoothed anomalies was performed, and the first two SSTA EOFs are shown in Figs. 1a and 1c. The principal component (PC) time series corresponding to the first and second EOFs are shown in Figs. 1b and 1d. The first EOF pattern, which accounted for 61.33% of the total SSTA variance, represented the mature ENSO phase (El Niño or La Niña), and the corresponding PC time series was highly correlated (with a correlation coefficient of



0.85) with the cold tongue index (SST anomaly averaged over 4 °S–4 °N, 180 °–90 °
W) over the whole period.   The second EOF, accounting for 14.52% of the total
SSTA variance, indicated the ENSO signal beginning to decay. Compared with the
first mode, these were slightly attenuated in terms of the scope and intensity. The
above analysis is similar to the EOF analysis of the SSTA field in the previous studies
(Johnson et al., 2000;Timmermann et al., 2001). This indicates that the front two
variance contribution modes can describe the main characteristics of the SSTA field
and El Niño/La Niña. Therefore, we can choose the $T_1, T_2$ time series EOF
decomposition modes as the modelling objects.

## 2.3 Selection of other prediction model factors

The ENSO intensity impact factor is an important issue in the ENSO

prediction. Previous studies have found that teleconnection patterns, temperature,
precipitation, wind and SSH may affect the ENSO strength (Trenberth et al.,1998;
Webster,1999; Ashok et al., 2001; Yoon and Yeh, 2010; Tomita and Yasunari, 1996).
For example, Trenberth et al. (1998) noted that the Pacific North American Oscillation
Index (PNA) and SOI in the Pacific Intertropical Convergence Zone (ITCZ) were all
closely related to ENSO. Liao et al. (2007) also noted that the decadal variation
during ENSO events had a close relationship with the SOI index.The vast majority of
studies (Tomita and Yasunari, 1996; Zhou and Wu, 2010) have concentrated on the
impacts of ENSO on the East Asian winter monsoon (EAWM). During the EAWM
season, ENSO generally reaches its mature phase and has the most prominent impact
on the climate. Wang et al. (1999a) and Wang et al. (1999b) suggested that the zonal



wind factors in the eastern and western equatorial Pacific played a critical role in the
transition phase of the ENSO cycle, which could excite eastward propagating Kelvin
waves and affect the SSTA in the equatorial Pacific.

Based on the above analysis, we selected four factors, which may be closely

related with the ENSO index (Niño 3.4) and were obtained as follows:

(1) The zonal wind in the eastern equatorial Pacific factor (u1) was calculated

as the grid-point average of zonal wind in the area [5 °S ~ 5 °N, 150 °W ~ 90 °W].

(2) The PNA teleconnection factor was obtained from the CPC.

(3) The SOI factor was obtained from the CPC.

(4) The EAWM index (EAWMI) factor was proposed by Yang et al. (2002),

which is defined by the meridional 850-hPa winds averaged over the region (20 °
~40 °N, 100 °~140 °E).

All the four data selected ranged from January 1951 to January 2010.

Actually, how many variables and which variables are used in our model

become a key issue to be resolved. We can introduce a stepwise regression principle
to choose more reasonable predictors (Yim et al., 2015), because the stepwise
procedure can help selecting statistically important predictors at each step. The
significance of each predictor selected was based on its significance in increasing the
regressed variance by the standard $F$ test (Panofsky and Brier, 1968). A 95 %
statistical significance level was used as a criterion to select a new predictor at each
step. Once selected into the model, a predictor can only be removed if its significance
level falls below 95 % by the addition/removal of another variable. For example, for





the model of only one variable, because we forecast the ENSO index, we should
choose $T_1$ or $T_2$ as the variable. Considering that $T_1$ accounts for 61.33% of the total
SSTA variance, so we chose $T_1$ as the variable. For the model of two variables, there
are five factors ($T_2, u_1$, PNA, SOI and EAWMI) which can be chosen for the second
variable. Taking advantage of the stepwise regression ideas and selecting statistically
important predictors by a standard F test, we can find the largest F test value among
the five factors. That is $T_2$. Continuing this step, we can also select the reasonable
factors for the model of three variables. Based on this thought, when the number of
variables is determined, we can choose the most statistically important variables to
reconstruct the prediction model. The forecast results of these models can be seen in
table 1.

From table 1, the forecast results of all six models are satisfactory, where the

temporal correlations of the models are all greater than 0.60 and the root mean square
errors are all less than 0.81. Among all six models, the forecast results of four
variables are the best for the following reasons:

(1) In general, the amount of parameters is less than 10% of the sample size,

which can avoid over-fitting (Tetko et al., 1995) .The number of parameters
$a_1, a_2, ...a_{14}, b_1, b_2, ...b_{14}, c_1, c_2, ...c_{14}, d_1, d_2, ...d_{14}$ of the model of four variables $T_1, T_2, SOI, EAWMI$ is 56,
but we deleted the parameters which contributed little to the prediction. That means
that there are 56 parameters in equation (1) in section 3, but there are only 34
parameters in equation (3) in section 3which is our final prediction equation. In
section 5.1, because $p$ is identified as 6, the number of parameters of the




self-memorization function $\beta_i$ is 28. Therefore, the total number of parameters in the
model of four variables is 62, which is less than 10% of the sample size (720 months).
The number of parameters $a_1, a_2, ... a_{20}, b_1, b_2, ... b_{20}, c_1, c_2, ... c_{20}, d_1, d_2, ... d_{20}, e_1, e_2, ... e_{20}$ of the model
of five variables $T_1, T_2, SOI, EAWMI, u_1$ is 100. Although the parameters which contributed a
little were deleted, the number was still 72, and the number of self-memorization
parameters was 30 ( $p$  determined as 5). Thus, the total number of parameters in the
model of five variables was 102, which was more than 10% of the sample size (720
months).This will cause an overfitting problem. Hence, when we selected the model
of five or six variables which entailed large amounts of computation that made
precision difficult, and too many parameters caused an overfitting phenomenon. That
is why the forecast results of five or six variables were worse than those of four
variables.

(2) The models of one, two and three variables can avoid the overfitting problem,

but too few variables will result in too few reconstruction parameters, causing
important information missing from the model. Especially, when the model of one or
two variables was considered, we only studied the self-memorization of the ENSO
system but did not consider the mutual-memorization between factors. Thus,
equations of our model only contained a self-memory term, not an exogenous effect
term. That is why the forecast results of one, two and three variables were worse than
those of four variables.

Based on the above analysis, we finally chose  $T_1$,  $T_2$, SOI and EAWMI as

predictors for the model.



## 3. Reconstruction of dynamical model based on GA


Takens' delay embedding theorem (Takens, 1981) provides the conditions under
which a smooth attractor can be constructed from observations made with a generic
function. Later results replaced the smooth attractor with a set of arbitrary
box-counting dimensions and the class of generic functions with other classes of
functions. Takens had shown that if we measured any single variable with sufficient
accuracy for a long period of time, it would be possible to construct the underlying
dynamical structure of the entire system from the behavior of that single variable
using delay coordinates and the embedding procedure. It was therefore possible to
construct a dynamical model of system evolution from the observed time series.
Introducing this idea here, four time series of the $T_1$, $T_2$, SOI and EAWMI factors
were chosen to construct the dynamical model.
The basic idea of statistical-dynamical model construction is discussed in
Appendix A and was introduced in our previous work (Zhang et al., 2006; Hong et al.,

2014).

A simplified second-order nonlinear dynamical model can be used to depict the
basic characteristics of atmosphere and ocean interactions (Fraedrich, 1987). Suppose
that the following nonlinear second-order ordinary differential equations are taken as
the dynamical model of reconstruction. In the equations, $x_1, x_2, x_3, x_4$ were used to
represent the time coefficient series of $T_1$, $T_2$, SOI and EAWMI.





$$\frac{dx_1}{dt} = a_1 x_1 + a_2 x_2 + a_3 x_3 + a_4 x_4 + a_5 x_1^2 + a_6 x_2^2 + a_7 x_3^2 + a_8 x_4^2 + a_9 x_1 x_2 + a_{10} x_1 x_3 + a_{11} x_1 x_4 + a_{12} x_2 x_3 + a_{13} x_2 x_4 + a_{14} x_3 x_4$$

$$\frac{dx_2}{dt} = b_1 x_1 + b_2 x_2 + b_3 x_3 + b_4 x_4 + b_5 x_1^2 + b_6 x_2^2 + b_7 x_3^2 + b_8 x_4^2 + b_9 x_1 x_2 + b_{10} x_1 x_3 + b_{11} x_1 x_4 + b_{12} x_2 x_3 + b_{13} x_2 x_4 + b_{14} x_3 x_4$$


$$\frac{dx_3}{dt} = c_1 x_1 + c_2 x_2 + c_3 x_3 + c_4 x_4 + c_5 x_1^2 + c_6 x_2^2 + c_7 x_3^2 + c_8 x_4^2 + c_9 x_1 x_2 + c_{10} x_1 x_3 + c_{11} x_1 x_4 + c_{12} x_2 x_3 + c_{13} x_2 x_4 + c_{14} x_3 x_4$$

$$\frac{dx_4}{dt} = d_1 x_1 + d_2 x_2 + d_3 x_3 + d_4 x_4 + d_5 x_1^2 + d_6 x_2^2 + d_7 x_3^2 + d_8 x_4^2 + d_9 x_1 x_2 + d_{10} x_1 x_3 + d_{11} x_1 x_4 + d_{12} x_2 x_3 + d_{13} x_2 x_4 + d_{14} x_3 x_4$$


$\hspace{12cm}$ (1)
$\hspace{1cm}$ Based on the parameter optimization search method of GA in Appendix A, the
time coefficient series of $T_1$, $T_2$, SOI and EAWMI from January 1951 to April 2008
are chosen as the expected data to optimize and retrieve model parameters. To avoid
the overfitting problem, we used $\quad x_{nor} = \dfrac{x - x_{min}}{x_{max} - x_{min}}\quad$ to normalize the raw value of
each of the four predictors, then we used the normalized value to model and forecast.
Finally, we made forecast results revert back to the raw data magnitude by
$x = x_{nor}(x_{max} - x_{min}) + x_{min}$ .
$\hspace{1cm}$ After eliminating weak items with small dimension coefficients, the nonlinear
dynamical model of the first time series $T_1$, the second time series $T_2$, SOI and EAWMI
can be reconstructed as follows:

$$\frac{dx_1}{dt} = F_1 = -0.3328 x_1 + 1.2574 x_2 - 0.3511 x_3 - 0.0289 x_1^2 + 3.1280 x_3^2 + 0.0125 x_1 x_2 + 2.7805 x_1 x_3 - 1.5408 x_2 x_4$$

$$\frac{dx_2}{dt} = F_2 = 1.0307 x_1 - 3.1428 x_2 + 0.3095 x_4 + 4.2301 x_1^2 - 1.2066 x_2^2 + 2.5024 x_4^2 - 0.2891 x_1 x_3 + 0.7815 x_1 x_4 - 0.4266 x_3 x_4$$


$$\frac{dx_3}{dt} = F_3 = -2.3155 x_1 + 3.2166 x_3 + 1.5284 x_4 - 1.4527 x_2^2 - 0.0034 x_3^2 - 4.1206 x_4^2 - 0.0025 x_1 x_4 + 0.0277 x_2 x_3 + 1.2860 x_2 x_4$$

$$\frac{dx_4}{dt} = F_4 = 0.4478 x_2 - 0.0268 x_4 + 0.8995 x_1^2 - 2.3890 x_3^2 + 0.2037 x_4^2 + 1.3035 x_1 x_2 + 2.0458 x_1 x_4 - 2.0015 x_2 x_4$$

$\hspace{12cm}$ (2)





The appropriate model coefficient estimates determine the robustness of the
model and the accuracy of forecast results. We should now judge whether the model
coefficients are appropriate or not.
Frist, the largest Lyapunov exponent (LLE) is one of the indexes that can
represent the characteristics of chaotic systems. The final Lyapunov exponents of Eq.
(2) were [0.0433,0.0012,-0.1285], containing both a negative Lyapunov exponent
and two positive Lyapunov exponents, which demonstrate that our dynamic system is
indeed a chaotic system.
Second, we calculated the equilibrium roots of Eq. (2). Only the third
equilibrium was adjudged to be stable, based upon higher-order terms within the
Taylor series, the indices of which were mostly in accordance with the actual weather
system. The indices in the unstable equilibria could not accurately describe the actual
weather. Based on these two aspects, we can see that the model coefficient estimates
were reasonable and reflected the dynamical characteristics of the model.
The model required testing. Because the training period was from January 1951
to April 2008, we chose $T_1$, $T_2$,SOI and EAWMI of May 2008, which were not used
as initial forecast data in the modeling. Next, the Runge–Kutta method was used to do
the numerical integration of the above equations, and every step of the integration was
regarded as 1 month's worth of forecasting results. As a result, forecast results of four
time series over a period of 20 months were obtained. Here, the focus was on the
forecast results of $T_1$ and $T_2$, as shown in Fig.2.
From Fig. 2, forecast performance of $T_1$ and $T_2$ within 5 months was better.



Using $T_1$ as an example, at this time, the temporal correlation between model
predictions and corresponding observations was 0.8966 and the mean absolute
percentage error (MAPE) (Hu et al., 2001), $MAPE = \dfrac{1}{n}\sum_{i=1}^{n}\left|\dfrac{D_e(i)-D_0(i)}{D_0(i)}\right|\times 100$, was
8.32%. However, after 5 months, MAPE increased rapidly, and was 31.29% at 10
months. The model forecast then significantly diverged from observations, and the
forecast became inaccurate. After 10 months, the forecast results became increasingly
worse, which indicated that the forecast of the model after 5 months was unacceptable.
The forecast results of $T_2$ were similar to those of $T_1$.

The model's skill should be further assessed by cross-validated retroactive

hindcasts of the time series. As in the above example, omitting a portion of the time
series (12 months, January 1951 to January 1952) from observations, we trained the
model based on the data from February 1952 to December 2010, and then predicted
the omitted segments (12 months, January 1951 to January 1952). We then repeated
this procedure by moving the omitted segment along the entirety of the available time
series. Each experiment have used the different training sample and have established
the different model equation (but the method is the same).Finally, we obtained
cross-validated retroactive hindcast results of $T_1$ and $T_2$, as shown in Fig. 3. Figure
3 is combined results of the 60 forecast experiments.

As Fig. 2, the forecast performance of $T_1$ and $T_2$ in Fig. 3 was not satisfactory.

The model forecast significantly diverged from observations, and the forecast became
inaccurate. The temporal correlations of $T_1$ and $T_2$ between model predictions and
corresponding observations were 0.3411 and 0.4176, respectively. Additionally, the





mean absolute percentage errors (MAPE) of $T_1$ and $T_2$ were 65.42% and 57.56%,
respectively. This indicates that the forecast of the model in the long -term was
inaccurate and unacceptable.

The forecast result may be inaccurate when the integral forecasting time is long.

There will be a significant divergence which will cause an ineffective forecast. To
improve the forecast accuracy, the forecast not only depends on the integral equation
but also on a single initial value. Choosing the different initial value will cause
different forecast accuracy. For example, in a total of 60 cross-validated retroactive
hindcasts examples, the minimum MAPE was 37.65%, while the maximum MAPE
was 89.88%. A forecast, depending on a single initial value, will cause instability of
the forecast results. These two problems are addressed by introducing the
self-memorization principle in the next section.

## 4. Introduction of self-memorization dynamics to improve the reconstructed model

In the above discussion, it was shown that the accuracy of the forecast results of

equation (2) were unsatisfactory. To improve long-term forecasting results, the
principle of self-memorization can be introduced into the mature model (Gu 1998;
Chen et al., 2009). The principle of self-memorization dynamics (Cao, 1993; Feng et
al., 2001) can be seen in Appendix B.

Based on Eq. (B10) in Appendix B, the improved model can be expressed as



$$\text{follows:} \begin{cases} x_{1t} = \sum_{i=-p-1}^{-1} \alpha_{1i} y_{1i} + \sum_{i=-p}^{0} \theta_{1i} F_1(x_{1i}, x_{2i}, x_{3i}, x_{4i}) \\ x_{2t} = \sum_{i=-p-1}^{-1} \alpha_{2i} y_{2i} + \sum_{i=-p}^{0} \theta_{2i} F_2(x_{1i}, x_{2i}, x_{3i}, x_{4i}) \\ x_{3t} = \sum_{i=-p-1}^{-1} \alpha_{3i} y_{3i} + \sum_{i=-p}^{0} \theta_{3i} F_3(x_{1i}, x_{2i}, x_{3i}, x_{4i}) \\ x_{4t} = \sum_{i=-p-1}^{-1} \alpha_{4i} y_{4i} + \sum_{i=-p}^{0} \theta_{4i} F_4(x_{1i}, x_{2i}, x_{3i}, x_{4i}) \end{cases} \quad (3)$$


where $y_i$ is replaced by the mean of two values at adjoining times; i.e.,
$y_i \equiv \dfrac{1}{2}(x_{i+1} + x_i)$; $F$ is the dynamic core of the self-memorization equation, which
can be obtained from Eq. (2); and $\alpha$ and $\theta$ are the memory coefficients, the formula
for which can be found in Appendix B.
If the values of $\alpha$ and $\theta$ can be obtained, Eq. (3) can be used to obtain the
results of final prediction. The memory coefficients $\alpha$ and $\theta$ in Eq. (3) were
calibrated using the least-squares method with the same data (January 1951 to April
2008) as those used in Section 3. Eq. (3) can be deconstructed as follows (M is the
length of the time series):
$$X = \begin{bmatrix} x_{11} \\ x_{12} \\ \cdot \\ \cdot \\ \cdot \\ x_{1M} \end{bmatrix}, \alpha = \begin{bmatrix} \alpha_{-p-1} \\ \alpha_{-p} \\ \cdot \\ \cdot \\ \cdot \\ \alpha_{-1} \end{bmatrix}, Y = \begin{bmatrix} y_{-p-1,1} & y_{-p,1} & \cdots & y_{-1,1} \\ y_{-p-1,2} & y_{-p,2} & \cdots & y_{-1,2} \\ \cdot & \cdot & & \cdot \\ \cdot & \cdot & & \cdot \\ \cdot & \cdot & & \cdot \\ y_{-p-1,M} & y_{-p,M} & \cdots & y_{-1,M} \end{bmatrix}, \Theta = \begin{bmatrix} \theta_{-p} \\ \theta_{-p+1} \\ \cdot \\ \cdot \\ \cdot \\ \theta_0 \end{bmatrix},$$

$$F = \begin{bmatrix} F_{-p,1} & F_{-p+1,1} & \cdots & F_{0,1} \\ F_{-p,2} & F_{-p+1,2} & \cdots & F_{0,2} \\ \cdot & \cdot & & \cdot \\ \cdot & \cdot & & \cdot \\ \cdot & \cdot & & \cdot \\ F_{-p,M} & F_{-p+1,M} & \cdots & F_{0,M} \end{bmatrix}$$



The matrix equation is:

$$X = Y\alpha + F\theta \qquad (4)$$

where $Z = [Y \vdots F], \quad W = \begin{bmatrix} \alpha \\ \vdots \\ \Theta \end{bmatrix}$ .
Eq. (4) can be written as:

$$X = ZW \qquad (5)$$

The memory coefficients vector $W$ can be calibrated using the least squares
method:

$$W = (Z^T Z)^{-1} Z^T X \qquad (6)$$

The memory coefficients $a, \theta$ can be obtained from Eq. (6). We then made a
prediction using the self- memorization equation (3), which used the $p$ values before
$t_0$ .
The coefficients in F and W were used with the same training data from January
1951to Apr.il 2008. In the forecast examples, we trained both the coefficients in F and
W at the same time, but in the paper we describe them separately to facilitate the
reader for better understanding.
**5. Model prediction experiments**
**5.1 Forecast of time series $T_1$ and $T_2$**
The training sample for the model was from January1951 to April 2008. Here, from
Eq. (3), the forecast results using $T_1, T_2$, SOI and EAWMI factors can be calculated, called
as step-by-step forecast.
When the retrospective order $p$ is confirmed, step-by-step forecasts can be



carried out. For example, when the $T_1, T_2$, SOI and EAWMI values of May 2008 were
forecast, $y_i$ was obtained from the previous $p + 1$ time of $T_1, T_2$, the SOI and the
EAWMI data, and $F_i(x_{1i}, x_{2i}, x_{3i}, x_{4i})$ was obtained from the previous $p$ times of
$T_1, T_2$, the SOI and the EAWMI data. All four equations were integrated simultaneously.
Taking these in Eq. (3), we can get the $T_1, T_2$, SOI and EAWMI values of May 2008,
which these can be taken as the initial values for the next prediction step. Then, the
$T_1, T_2$, SOI and EAWMI values from June 2008 and so on, can be generated.
5.1.1 Determination of $p$

Based on the self-memorization principle, the self-memorization of the system

determines the retrospective order $p$ (Cao, 1993). If the system forgets slowly,
parameters $a$ and $\theta$ will be small and the $p$ value should be high. The SSTA field
forecasts were on a monthly scale, the change of which was slow in contrast to
large-scale atmospheric motion. So parameters $a$ and $\theta$ were small, and generally,
the $p$ value was in the range 5 to 15.

The retrospective order $p$ was obtained by a trial calculation method. We selected

the $p$ values in the range 4 to 16 to construct the model. The correlation coefficients
(CC) and MAPE of long-term fitting test (from February 1951 to December 2010) are
shown in Table 2, which can be used as the standard to determine the retrospective
order $p$.

Table 2 indicates that when $p = 6$, the MAPE values of long-term fitting test

were the smallest and the correlation coefficients were the largest. Also, when $p$ from
5 to 9, CCs were all more than 0.58 and the forecast results were all good, which is





consistent with our interpretation of the physical mechanisms in section 6.2 below.
SOI and EMWMI were 5-12 months lead relationships with SST (Xu et al., 1993;
Chen et al, 2010; Wang et al., 2003). Using a cumulative period of SOI , EMWMI 5-8
months ahead as initial values can help improve the final forecast results. Our results
in table 2 are consistent with the actual physical ENSO process. Therefore, we
selected the retrospective order as $p$=6.
Then, the prediction experiments can be carried out, based on improved
self-memorization Eq. (3).
The improved self-memorization equation of $T_1, T_2$, SOI and EAWMI can then be
established. After the differential equation was discretely dealt with, the memory
coefficients were solved by the least-squares method given in section 4 (Training
period is January 1951 to April 2008). Finally, the improved prediction equation of
$T_1, T_2$, SOI and EAWMI, based on the self-memorization principle, can be expressed
as:

$$
\begin{cases}
x_{1t} = \sum_{i=-7}^{-1} \alpha_{1i} y_{1i} + \sum_{i=-6}^{0} \theta_{1i} F_1(x_{1i}, x_{2i}, x_{3i}, x_{4i}) \\
x_{2t} = \sum_{i=-7}^{-1} \alpha_{2i} y_{2i} + \sum_{i=-6}^{0} \theta_{2i} F_2(x_{1i}, x_{2i}, x_{3i}, x_{4i}) \\
x_{3t} = \sum_{i=-7}^{-1} \alpha_{3i} y_{3i} + \sum_{i=-6}^{0} \theta_{3i} F_3(x_{1i}, x_{2i}, x_{3i}, x_{4i}) \\
x_{4t} = \sum_{i=-7}^{-1} \alpha_{4i} y_{4i} + \sum_{i=-6}^{0} \theta_{4i} F_4(x_{1i}, x_{2i}, x_{3i}, x_{4i})
\end{cases}
\quad (7)
$$

where





$$\alpha=[\alpha_{ij}]=\begin{bmatrix} 0.0315 & -2.113 & 0.0284 & 2.1468 & 0.0688 & -0.7014 & 1.3248 \\ 0.4088 & -1.887 & -1.0233 & 1.5485 & 0.9028 & 1.0255 & -0.6443 \\ -0.9088 & -0.2557 & 0.9671 & -0.0054 & 1.0568 & 2.9764 & -0.5234 \\ 0.2088 & -1.0567 & 0.4891 & -0.5066 & -0.4890 & 1.4555 & 1.0966 \end{bmatrix}$$

$(i=0,1,...,4; j=-7,-6,...,-1)$

$$\theta=[\theta_{ij}]=\begin{bmatrix} 0.0485 & 0.0425 & -1.7688 & 0.8543 & 2.8901 & -0.1788 & -0.9066 \\ 0.07642 & 0.0941 & -1.2466 & -0.2288 & 0.1097 & 2.3221 & -1.4228 \\ -0.5288 & 1.2368 & -0.5568 & -0.0155 & 0.2886 & -0.1560 & 1.2775 \\ 1.5335 & -0.2887 & -0.5336 & -0.6072 & -0.5611 & 1.0225 & -1.0625 \end{bmatrix}$$

$(i=0,1,...,4; j=-6,-5,...,0)$

The step-by-step forecast was performed. The retrospective order $p=6$ means that earlier seven observation data ($p+1=7$) should be used during the forecasting process. The forecast results per month were saved for the next period predictions.

5.1.2 Long-term step-by-step forecasts of $T_1$ and $T_2$

To test the actual forecast performance of the above improved model, long-term step-by-step forecasts of $T_1$ and $T_2$ from May 2008 to December 2010 for 20 months were carried out, as shown in Fig. 4. The forecast results of $T_1$ and $T_2$ were good. Within 8 months, the correlation coefficients of $T_1$ and $T_2$ were 0.9163 and 0.9187. MAPEs of $T_1$ and $T_2$ were small, only 5.86% and 6.78%. The forecast time series from 8 months to 14 months gradually diverged, but the trend was acceptable. The correlation coefficients of $T_1$ and $T_2$ reached 0.8375 and 0.8251, and MAPEs of $T_1$ and $T_2$ were 8.32% and 9.11%. After 14 months, forecast began to diverge and the error started to increase, but the correlation coefficients of $T_1$ and $T_2$ remained about 0.6899 and 0.6782, and MAPEs reached 18.31% and 19.44%, which can be acceptable.

**5.2 Cross-validated retroactive hindcasts of time series $T_1$ and $T_2$**





As in section 3, the model's skill should be further assessed by cross-validated
retroactive hindcasts of the time series. Because our step-by-step forecasts need the
earlier seven observation data ( $p + 1 = 7$ ), we can obtain cross-validated retroactive
hindcast results of $T_1$ and $T_2$ from August 1951 to December 2010, as shown in Fig.

5.

From Fig. 5, the forecast performance of $T_1$ and $T_2$ was good. The

correlation coefficients of $T_1$ and $T_2$ were 0.7124 and 0.7036, respectively. The
MAPEs of $T_1$ and $T_2$ were small, only 19.57% and 19.79%, respectively. The peaks
and valleys of $T_1$ and $T_2$ were also forecasted accurately. The forecast results
indicated that the cross-validated retroactive hindcast results of $T_1$ and $T_2$ were close
to the observed values. Compared to Fig. 3, the improved model had better forecast
abilities than the original model.

Many researchers (Zhang et al., 2003b; Smith, 2004) have used Oceanic Niño

Index (ONI) which is used by the U.S. NOAA Climate Prediction Center to determine
the El Niño and La Niña years. It defined that the ONIs of five consecutive months in
winter were all more than 0.5 (less than -0.5) is the ElNiño (La Niña) year. Based on
the above criterion, we can divide the total 60 years (1951-2010) into three categories.
It includes the 18 examples of ElNiño year (such as 1958, 1964, 1966, etc.), 22
examples of LaNiña year (such as 1951, 1955, 1956, etc.) and the remaining 20
experiments of the neutral year. Since the details in Fig.5 is not clear, we list the
forecast results of 60 experiments (including 18 El Niño examples, 22 La Niña
examples and 20 Neutral examples) in table 3.



From table 3, the average of CC of both $T_1$ and $T_2$ of 60 experiments within
6 months was more than 0.84 and MAPE was less than 8%. The average of CC within
12 months was more than 0.74 and MAPE was less than 12%. According to the
literature (Barranel et al., 1999), when MAPE was less than 15%, which means the
error was not great and the forecast results were good. Obviously, the forecast results
of ElNiño / LaNiña experiments were a little worse than those of neutral examples,
which means the forecast ability of our model for the abnormal situation was a little
worse than those for the normal situation. But even for ElNiño / LaNiña experiments,
the average of CC was still more than 0.7 and MAPE was less than 15%, which
means the error was not too large and was still within an acceptable range.
**5.3 Forecast of the SSTA field**
When we obtained the forecast results of the time coefficient series $T_1$ and $T_2$,
we submitted them into the following equation to reconstruct the forecast SSTA field:
$$\hat{x}_t = \sum_{n=1}^{2} E_n \bullet T_{nt}, t = 1, 2, ..., 12 \qquad (8)$$
where $E_n$, $T_{nt}$ are the EOF space fields and forecast time coefficients,
respectively, and $\hat{x}_{tj}$ is the forecast SSTA field reconstructed by EOF.
After reconstruction of the space mode (treated as constant) and time coefficient
series (model prediction), the forecast of the SSTA fields was obtained, based on the
forecast results of $T_1$ and $T_2$ in Section 5.2. For economy of space, we cannot draw
all of the forecasted SSTA fields, so we selected a strong El Niño event (December
1997), a strong La Niña event (December 1999) and a neutral event (November 2002)
as examples.
Fig. 6 shows the forecast SSTA field during a strong El Niño event. From the


actual SSTA field in December 1997 (Fig. 6a), an obvious warm tongue structure
occurred in the area of [10 ̊S～5 ̊N,90 ̊W～150 ̊W] in the Eastern Equatorial Pacific,
and a warm anomalous distribution arose in the west Pacific, which indicated a weak
El Niño event. The forecasted SSTA field of December 1997 is shown in Fig. 6b.
Although the range of warm tongue was a litter bigger than the actual situation, the
forecast shape was similar to the actual field and also the contour lines were similar.
The average MAPE between the forecast field and the actual field is 8.56%, which
was controlled within 10%. The forecast results of the improved model event were
quite good for the El Niño event.
Fig.7 shows the forecasted SSTA field of a strong La Niña event. From the actual
SSTA field in December 1999 (Fig. 7a), an obvious cold pool occurred in the area of
[10 ̊S～10 ̊N,120 ̊W～180 ̊W] in the Equatorial Pacific, which covered the Niño3.4
area. This SSTA field presented a strong strength La Niña event. The forecast SSTA
field from December 1999 is shown as Fig. 7b. Although the strength of the cold pool
was weaker than the actual situation, the forecast shape was similar to that of the
actual field. The average MAPE between the forecast field and the actual field was
9.69%. The errors were larger than that of the El Niño event, but they can be
controlled within 10%, which is acceptable.
Fig. 8 shows the forecasted SSTA field of a neutral event. From the actual SSTA
field in November 2002 (Fig. 8a), a warm pool occurred in the area of [10 ̊S～10 ̊N,
120 ̊W～180 ̊W] in the Equatorial Pacific, which covered the Niño3.4 area. However,
the warm pool was small and weak, which represented a neutral event. The forecasted



SSTA field from November 2002 is shown in Fig. 8b. Comparing Figures 6, 7 and 8,
we can see that the forecasted SSTA field of a neutral event was a little worse than
thatof the El Niño and La Niña events. The forecasted shape of the SSTA field
basically described the actual situation, but the warm pool in the Niño3.4 area was
stronger and bigger than that of the actual situation, which indicated a borderline El
Niño event. The average MAPE between the forecasted field and the actual field was
14.50%, which was big but can be accepted.

We obtained the average values of MAPE of 18 El Niño events, 22 La Niña

events and 20 neutral events, which were 9.52%, 9.88% and 14.67%, respectively,
representing a good SSTA field forecasting ability of our model.
**5.4 Forecast of ENSO index**

The ENSO index can be represented as the sea surface temperature anomaly

(SSTA) in the Niño-3.4 region (5 °N-5 °S, 120 °-170 °W) and the ENSO index
forecast was the 3-month forecast (Barnston et al. 2012). So we also can pick up the
ENSO index from the above forecasted SSTA field. The forecast results of the ENSO
index within 20 months can also be obtained. The definition of lead time can be seen
in the reference (Barnston et al. 2012). Therefore, similar to the forecast experiment in
section 5.1, a succession of running 3-month mean SST anomalies with respect to the
climatological means for the respective prediction periods, averaged over the Niño 3.4
region, can be obtained, as demonstrated in Fig. 9.

The forecast results within lead times of 18 months are shown in Fig. 9, which

demonstrate that the forecast results of the ENSO index are good. Within lead time of



12 months, the correlation coefficient was 0.8985 and the MAPE value was small,
only 8.91%. In addition, the borderline La Niña event in 2008–2009 was predicted
well. After lead times of 12 months, forecasts began to diverge and the errors started
to increase. Although the correlation coefficient remained approximately 0.61, MAPE
reached 18.58%. Therefore, a moderate strength El Niño event that occurred in
2009/10 was not predicted.
We should give more examples to test the ENSO prediction ability of our model.
As in section 5.3, we can divide 60 examples as three types, which are examples of
ElNiño year, LaNiña year and neutral year. Finally, we can obtain the forecast results
of different types of examples in different lead times, as shown in table 4.
From table 4, the average CC of 60 experiments was 0.712 and the average
MAPE was 7.62% within 12 months for all seasons of lead time, which indicates that
the overall ENSO forecast ability of our model was good. The forecast results of the
El Niño examples were significantly worse than those of La Niña examples, while the
forecast results of La Niña examples were significantly worse than those of neutral
examples, which show the model forecast ability of the abnormal state was worse than
the normal state of the ENSO index. Even for the forecast results of El Niño examples,
the average CC was still above 0.6 and the average MAPE can be controlled below
10%, which means the forecast results were still in the acceptable range. Our model
not only accurately predicted the stronger El Niño and La Niña phases but also the
neutral states. But the forecast results in summer were a little worse than those in
winter, as shown in Fig.10.



The ENSO forecast often had a spring predictability barrier (Webster, 1999),
which was most prominent during decades of relatively poor predictability
(Balmaseda et al., 1995).To test our model, the skill should be computed over the
entire time series and separately for seasonal subsets of the time series. The average
cumulative correlation coefficient and MAPE of winter were compared with those of
summer, as shown in Fig.10. The average cumulative correlation and average
cumulative MAPE values between the forecast values and the actual values changed
with time, from which good trends of forecast results can be seen. As long as the
forecast time increased, the cumulative MAPE increased and the correlation decayed
gradually. The forecast results appeared to diverge. Although the forecast results of
the present model in the summer were worse than in the winter, the margin was not
high, which means the model can overcome the "spring predictability barrier," to
some extent.
**5.5 Compared with six mature models**
Barnston et al. (2012) compared many ENSO forecast models. Based on his
research, we selected four high quality dynamical models, including ECMWF, JMA,
the National Aeronautics and Space Administration Global Modelling and
Assimilation Office (NASAGMAO) and the National Centre for Environmental
Prediction Climate Forecast System (NCEP CFS; Version1). Two high quality
statistical models also be selected, including the University of California, Los Angeles
Theoretical Climate Dynamics (UCLA-TCD) multilevel regression model and the
NOAA/NCEP/CPC constructed Analogue (CA) model. The detail of the above

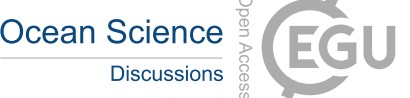

models can be seen in these references (Reynoldset al., 2002; Luo et al., 2005;
Barnston et al., 2012).

We then compared the forecast ability of the above six models with that of our

mode. All of the experiments of our model and six other models were conducted
under the same conditions using the same historical data for modelling and the same
initial values to forecast. In the CPC website, there are detailed explanations of six
models' training samples and the initial values. So we do not need to install all these
models on their own machines and run them for forecasting. We just made training
samples and initial values of our model were the same with those of selected six
models. At an 8-month lead time, the correlation ability of our model for all seasons
combined was 0.613 (Fig. 11). In brief, the forecast ability of the ECMWF model was
slightly better than that of our model but the ability of the other 5 models was worse
than that of our model. While, in regard to the forecast length, the temporal
correlation within 12 months of our model is greater than 0.6, which was superior to
the ECMWF model. In addition, the forecast results of the UCLA-TCD model and the
CPC CA model reduced quickly after 5-month lead times, so the forecast ability of
our model was more stable than them.

The root mean square error (RMSE) was also examined to assess the

performance of discrimination and calibration. Barnston et al. (2012) believed that all
seasonal RMSE values contributed equally to a seasonally combined RMSE. So we
drew figure 12 to show seasonally combined RMSE.

From Fig. 11 and Fig. 12, we can see the highest correlation tend to have



lower RMSE. So the RMSE of our model was slightly higher than that of ECMWF
model, but it was much lower than those of the other 5 models.
**6. Conclusions and discussion**
**6.1 Conclusions**

A new forecasting model of the SSTA field was proposed based on a dynamic

system reconstruction idea and the principle of self-memorization. The approach of
the present paper consisted of the following steps:

(1)    The SST field can be time (coefficients)-space (structure) deconstructed

using the EOF method. Take $T_1$, $T_2$, SOI and EAWMI and consider them as
trajectories of a set of four coupled quadratic differential equations based on the
dynamic system reconstruction idea. The parameters of this dynamic model were
estimated using a GA.

(2)    The forecast results of the dynamic model can be improved by the

self-memorization    principle.    The    memory    coefficients    in    the    improved
self-memorization model were obtained using the GA method.

(3)    The    long-term    step-by-step    forecast    results    and    cross-validated

retroactive hindcast results of time series $T_1$ and $T_2$ are all found to be good, with a
correlation coefficient of approximately 0.80 and a mean absolute percentage error of
less than 15%.

(4)    The    improved    model    was    used    to    forecast    the    SSTA    field.    The

forecasted SSTA fields of three types of events are accurate. Not only is the forecast
shape similar to the actual field but also the contour lines are similar.



(5)    The improved model was also used to forecast the ENSO index. The
average correlation coefficient of 60 examples within 12 months is 0.712, and the
MAPE value is small, only 7.62%, which proves that the improved model has better
forecasting results of the ENSO index. Although the forecast results of the model in
the summer were worse than in the winter, the margin was not high, which means that
the model can overcome the spring predictability barrier to some extent. Finally,
compared with the six mature models, the new dynamical-statistical forecasting
model has a scientific significance and practical value for the SST in the eastern
equatorial Pacific and El Niño/La Niña event predictions.

## 6.2 Discussion

Because the formula of our model includes a linear combination of 4 variables
($t_1, t_2$, SOI, EAWM), statistical forecasting requires independence between predictors.
We can calculate the correlation coefficients between variables, as shown in table 5.
In fact, as Table 5 shows, the correlation coefficients between the factors were all less
than 0.45, indicating the independence between factors. So this does not generate too
much redundancy and can avoid an overfitting problem, which can destroy the
stability of the model.
The introduction of self-memorization essentially introduces a lot of new
coefficients, which may cause an overfitting problem. Because we have selected a
model of four variables, there is a total of 62 parameters. In order to avoid the
overfitting problem, the sample sizes are more than 10% of the amount of parameters.
So our sample size is greater than 620 data to avoid the overfitting problem. If we

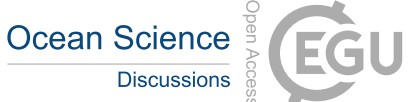

choose the model of three variables, the parameters in which will be less, the sample
size in this situation can be less. But the forecast results may be a little worse, based
on the analysis in section 2.3. So the length of training samples is related to the
number of parameters of our model.

Also, we have tried to detrend our data before the model constructed. But we

found the results didn't change too much. That is mean our model is not very
sensitive to climate change, so the detrended data has little effect for our model to
improve the forecast effect.

Compared with the original model, why the improved model has good forecast

results and can overcome the spring predictability barrier to some extent are as follow:
Recently, many studies have pointed out that spring is the most unstable season of the
air - sea interaction and the error is likely to develop or grow in the spring, resulting in
the spring predictability barrier (Zhang et al, 2012; Philander et al., 1992). When the
original model uses the indexes in summer as the initial values to predict, the SOI
factor representing the air-sea interaction is most unstable in the spring and the
EMWMI factor does not have much influence on ENSO in summer, so the forecast
results using the indexes in summer as the initial values are certainly much worse than
those using the indexes in the winter as the initial values. That is why our original
model does not overcome the spring predictability barrier.

However, the introduction of the self-memorization dynamics principle can help

our model overcome the spring predictability barrier to some extent. Although the
lead time is still summer (such as JJA), the information of the initial value actually



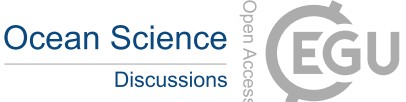

contains the previous p + 1 month (in this case p = 6, which contains the information
of the previous seven months, including the information of $T_1, T_2$  , SOI, EMWMI
factor in winter (January, February), spring (March, April, May) and summer (June
and July)). From the dynamical analysis, in this situation, the information and
interaction relationship of four factors have been a long period (from winter to
summer) accumulated, containing much air-sea interaction processes and winter
monsoon continued abnormal information, so the forecast results of our improved
model will be much better than the original model which simply uses only one initial
value. That is why the improved model overcomes the spring predictability barrier to
some extent.

The forecast results of our model are good, but it still has some problems:

(1) Although the reason why the improved model has good forecast results has

disucussed in the section6.2, the deep physical mechanisms that the proposed model
has dealt with is not very clear, so its dynamical characteristics should be further
analysed.

(2)The experiments in the present study have proven that the forecasting results

of the improved model are good for large-scale systems, such as ENSO events, and
the forecasting period has been extended. However, for small-scale systems, such as
Hurricanes, whether the forecast results could be improved using the present
improved model needs to be further verified.

(3) Our paper focuses primarily on these defined indices with $T_1, T_2$ to

reconstruct a prediction model. Maybe, we can select variables (predictor) based on





EOF analysis and our model may be a more physically oriented model. Maybe we can
learn from Yim et al (2013; 2015) to draw correlation maps between these fields and
the SSTA field and select the predictors from physical considerations. All these above
questions require that a lot of experiments to be carried out.

These items will be our future work.


**Acknowledgments** This study was supported by the Chinese National Natural
Science Fund (nos 41375002,41075045, 41306010, 41571017, 51190091 and
41071018) and the Chinese National Natural Science Fund (BK20161464) of Jiangsu
Province, the Program for New Century Excellent Talents in University
(NCET-12-0262), the China Doctoral Program of Higher Education
(20120091110026), the Qing Lan Project, the Skeleton Young Teachers Program, and
the Excellent Disciplines Leaders in Midlife-Youth Program of Nanjing University.

**APPENDIX A: THE PRINCIPLE OF DYNAMICAL MODEL**
**RECONSTRUCTION**

Suppose that the physical law of a nonlinear system going by over time can be

expressed as the following difference form:
$$\frac{q_i^{(j+1)\Delta t} - q_i^{(j-1)\Delta t}}{2\Delta t} = f_i(q_1^{j\Delta t}, q_2^{j\Delta t}, ...., q_i^{j\Delta t}, ...., q_N^{j\Delta t}) \quad j = 2,3,......M-1 \tag{A1}$$

where $f_i$ is the generalized nonlinear function of $q_1, q_2, ..., q_i, ..., q_N$, $N$ is the number
of variables, and $M$ is the length of observed data. $f_i(q_1^{j\Delta t}, q_2^{j\Delta t}, ...., q_i^{j\Delta t}, ...., q_N^{j\Delta t})$ can be assumed
to contain two parts: $G_{jk}$ representing the expanding items which contain variable




$q_i$ , $P_{ik}$ just representing the corresponding parameters which are real numbers
( $i = 1, 2, ... N$ , $j = 1, 2, ... M$ , $k = 1, 2, ..., K$ ).
It can be supposed as follows:
$$f_i(q_1, q_2, ..., q_n) = \sum_{k=1}^{K} G_{jk} P_{ik} \qquad (A2)$$

$D = GP$ is the matrix form of Eq.(A2) , in which
$$D = \begin{Bmatrix} d_1 \\ d_2 \\ ... \\ d_M \end{Bmatrix} = \begin{Bmatrix} \frac{q_i^{3\Delta t} - q_i^{\Delta t}}{2\Delta t} \\ \frac{q_i^{4\Delta t} - q_i^{2\Delta t}}{2\Delta t} \\ ... \\ \frac{q_i^{M\Delta t} - q_i^{(M-2)\Delta t}}{2\Delta t} \end{Bmatrix}, \quad G = \begin{Bmatrix} G_{11}, G_{12}, ..... G_{1K} \\ G_{21}, G_{22}, ..... G_{2,K} \\ ... \\ G_{M1}, G_{M2}, ..... G_{M,K} \end{Bmatrix}, \quad P = \begin{Bmatrix} P_{i1} \\ P_{i2} \\ ... \\ P_{iK} \end{Bmatrix} \qquad (A3)$$

Parameters of the above equation can be determined through inverting the
observed data. Vector P which satisfies the above equation can be solved, based on a
given vector D. Assuming $q$ is unknown, it is a nonlinear system. However, assuming
$P$ is unknown, it is a linear system.
With the restriction $S = (D - GP)^T (D - GP)$ as a minimum, GA is introduced as an
optimization solution search in the model parameters space.
Assuming that the parameters matrix $P$ is the population (solutions), the
$S = (D - GP)^T (D - GP)$ is an objective function, $l_i = \frac{1}{S}$ is the value of individual
fitness, and $L = \sum_{i=1}^{n} l_i$ is the value of total fitness. The operating steps of GA include:
creation and coding of initial population (solutions), fitness calculation, the choice of
male parents, crossover and variation, etc. A detailed theoretical explanation can be
got from Wang (2001). The step length is 1 month during the calculation. After
optimization searches and genetic operations, the target value can be rapidly
converged on and each optimal parameter of the dynamical equations can be obtained.





Through the above approach, we can obtain parameters of a nonlinear
dynamical system, and reconstruct the nonlinear dynamical equations from observed
data.

**APPENDIX     B:     THE     MATHEMATICAL     PRINCIPLE     OF**
**SELF-MEMORIZATION DYNAMICS OF SYSTEMS**
The dynamical equations of a system can be expressed as:
$$\frac{\partial x_i}{\partial t} = F_i(x, \lambda, t) \ i = 1, 2, ..., J \tag{B1}$$
where $J$ is an integer,$x_i$ is the $i$th variable of the system state, and $\lambda$ is
the parameter. Equation (B1) represents the relationship between a source function
$F$ and a local change of $x$. Obviously, $x$ is a scalar function with time $t$ and
space $r_0$. A set of time $T = [t_{-p}...t_0...t_q]$ can be considered, where $t_0$ is an initial
time. A set of space $R = [r_a...r_i...r_\beta]$ can be considered, where $r_i$ is a spatial point.
An inner product in space $L^2 : T \times R$ is defined by:
$$(f, g) = \int_a^b f(\xi)g(\xi)d\xi, f, g \in L^2 \tag{B2}$$
Accordingly, a norm can be defined as:
$$\|f\| = [\int_a^b (f(\xi)^2 d\xi]^{1/2}$$
For a completion $L^2$, it can become a Hilbert space $H$. A generalized one
in $H$ can be regarded as a solution of the multi-time model. By introducing a
memorization function $\beta(r, t)$, we can obtain:





$$\int_{t_0}^{t} \beta(\tau)\frac{\partial x}{\partial \tau}d\tau = \int_{t_0}^{t} \beta(\tau)F(x,\tau)d\tau \qquad \text{(B3)}$$

where $r$ in $\beta(r,t)$ can be dropped through fixing on the spatial point $r_0$. Suppose
that function $\beta(r,t)$ and variable $x$ etc. are all continuous, differentiable and
integrable, an integration by the left parts of Eq. (B3) can be made as:
$$\int_{t_0}^{t} \beta(\tau)\frac{\partial x}{\partial \tau}d\tau = \beta(t)x(t) - \beta(t_0)x(t_0) - \int_{t_0}^{t} x(\tau)\beta'(\tau)d\tau \qquad \text{(B4)}$$

where $\beta'(t) = \partial\beta(t)/\partial t$. The mean value theorem can be introduced into the third
term in Eq. (B4), the following equation can be obtained:
$$-\int_{t_0}^{t} x(\tau)\beta'(\tau)d\tau = -x^m(t_0)[\beta(t) - \beta(t_0)] \qquad \text{(B5)}$$

where $x^m(t_0) \equiv x(t_m), t_0 < t_m < t$. Substituting Eq. (B4) and Eq. (B5) in Eq. (B3) and
carrying out an algebraic operation, the following equation can be obtained:
$$x(t) = \frac{\beta(t_0)}{\beta(t)}x(t_0) + \frac{\beta(t) - \beta(t_0)}{\beta(t)}x^m(t_0) + \frac{1}{\beta(t)}\int_{t_0}^{t} \beta(\tau)F(x,\tau)d\tau \qquad \text{(B6)}$$

Because the $x$ value which is at initial time $t_0$ and middle time $t_m$, only on
the fixed point $r_0$ itself, relates to the first term and the second term in Eq. (B6),
they are be called as a self-memory term. Also, we can call the third term as an
exogenous effect, i.e., which is contributed by other spatial points.
Similarly as Eq. (B4), for multi-time $t_i$, $i = -p, -p+1..., t_0, t$, it gives
$$\int_{t_{-p}}^{t_{-p+1}} \beta(\tau)\frac{\partial x}{\partial \tau}d\tau + \int_{t_{-p+1}}^{t_{-p+2}} \beta(\tau)\frac{\partial x}{\partial \tau}d\tau + ... + \int_{t_0}^{t} \beta(\tau)\frac{\partial x}{\partial \tau}d\tau = \int_{t_{-p}}^{t} \beta(\tau)F(x,\tau)d\tau.$$

After the same term $\beta(t_i)x(t_i), i = -p+1, -p+2, ..., 0$ was eliminated, we
have



$$\beta(t)x(t) - \beta(t_{-p})x(t_{-p}) - \sum_{i=-p}^{0}[\beta(t_{i+1}) - \beta(t_i)]x^m(t_i) - \int_{t_{-p}}^{t}\beta(\tau)F(x,\tau)d\tau = 0 \quad (B7)$$

As a matter of convenience, we set $\beta_t \equiv \beta(t), \beta_0 \equiv \beta(t_0), x_t \equiv x(t), x_0 \equiv x(t_0)$; the
following text uses similar notations. Then, Eq. (B7) can be expressed as:
$$\beta_t x_t - \beta_{-p}x_{-p} - \sum_{i=-p}^{0}x_i^m(\beta_{i+1} - \beta_i) - \int_{t_{-p}}^{t}\beta(\tau)F(x,\tau)d\tau = 0 \qquad (B8)$$

Setting $x_{-p} \equiv x_{-p-1}^m, \beta_{-p-1} = 0$, the Eq. (B8) can be written as:
$$x_t = \frac{1}{\beta_t}\sum_{i=-p-1}^{0}x_i^m(\beta_{i+1} - \beta_i) + \frac{1}{\beta_t}\int_{t_{-p}}^{t}\beta(\tau)F(x,\tau)d\tau = S_1 + S_2 \qquad (B9)$$

$S_1$ is called as a self-memory term and $S_2$ is called as an exogenous effect term.
For the convenience of calculations, the above self-memorization equation can
be discretized. The differential by difference and the summation can replace the
integration in Eq. (B9), and the mean of two values which are at adjoining times; i.e.,
$x_i^m \approx \frac{1}{2}(x_{i+1} + x_i) \equiv y_i$ can simply replace $x_i^m$.
Taking an equal time interval $\Delta t_i = t_{i+1} - t_i = 1$ and incorporating $\beta_i$ and $\beta_t$,
we can obtain a discretized self-memorization equation as follows:
$$x_t = \sum_{i=-p-1}^{-1}\alpha_i y_i + \sum_{i=-p}^{0}\theta_i F(x,i) \qquad (B10)$$

where $F$ is the dynamic kernel of the self-memorization equation, $\alpha_i = \frac{(\beta_{i+1} - \beta_i)}{\beta_t}$;
$\theta_i = \frac{\beta_i}{\beta_t}$.
Based on Eq. (B10), the above technique performed computations and the
forecast can be called as a self-memorization principle.

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



**List of Figures:**

**Fig.1** The time series (a) and the spatial mode (b) of the first mode; the time series(c) and the spatial mode (d) of the second mode of the SSTA filed

**Fig. 2** Forecast results of the first time coefficient series (a) and the second time coefficient series (b) of the SSTA field by the original model

**Fig. 3.** The cross-validated retroactive hindcast results of the first time coefficient series (a) and the second time coefficient series (b) of the SSTA field by the original model

**Fig. 4.** Long-term step-by-step forecast results of the first time coefficient series (a) and the second time coefficient series (b) of the SSTA field by the improved model

**Fig. 5.** The cross-validated retroactive hindcast results of the first time coefficient series (a) and the second time coefficient series (b) of the SSTA field by the improved model

**Fig. 6.** The forecast SSTA field (a) and the actual SSTA field (b) of an El Niño event (Dec.1997)

**Fig. 7.** The forecast SSTA field (a) and the actual SSTA field (b) of a La Niña event (Dec.1999)

**Fig. 8.** The forecast SSTA field (a) and the actual SSTA field (b) of neutral event (Nov.2002)

**Fig. 9.** The improved dynamical-statistical model prediction of the ENSO index

**Fig.10.** The cumulative correlation coefficients (a) and cumulative mean absolute percentage error (b) changing with time of different lead times

**Fig. 11.** Temporal correlation between model forecasts and observations for all seasons combined, as a function of lead time. Each line highlights one model.

**Fig.12.** RMSE in standardized units, as a function of lead time for all seasons combined. Each line highlights one model.

**Table captions:**

**Table 1.** Forecast results of models of different variables

**Table 2.** The correlation coefficient (CC) and Mean absolute percentage error (MAPE) of long-term fitting test when the retrospective order $p$ is different

**Table3.** The forecast results of $T_1$ and $T_2$ in different examples within 6 and 12 months

**Table4**. Temporal correlation (CC) and the mean absolute percentage error (MAPE) between model forecasts and observations within 12 months for November–January   December–February, and January–March as lead time of winter and for May-July, June-August and July-Sep. as lead time of summer.

**Table 5.** The correlation coefficients among four factors





**Figure:**

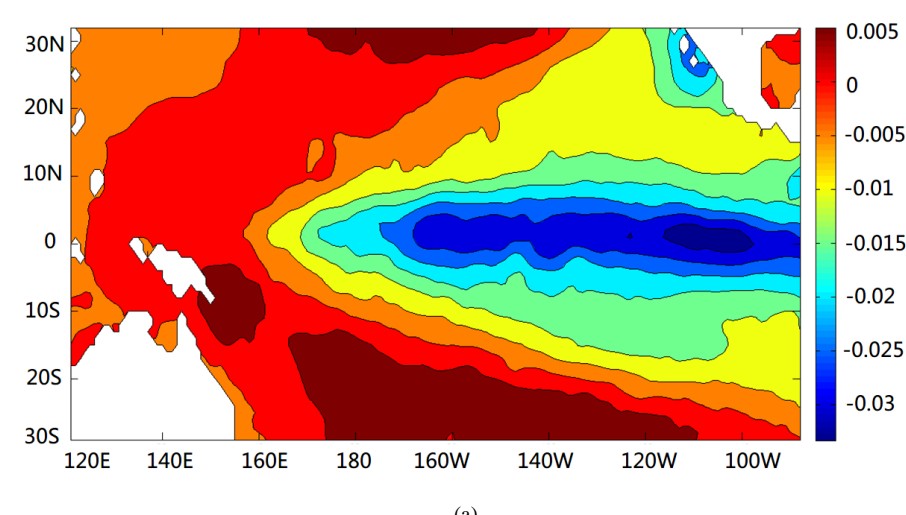


(a)


(b)



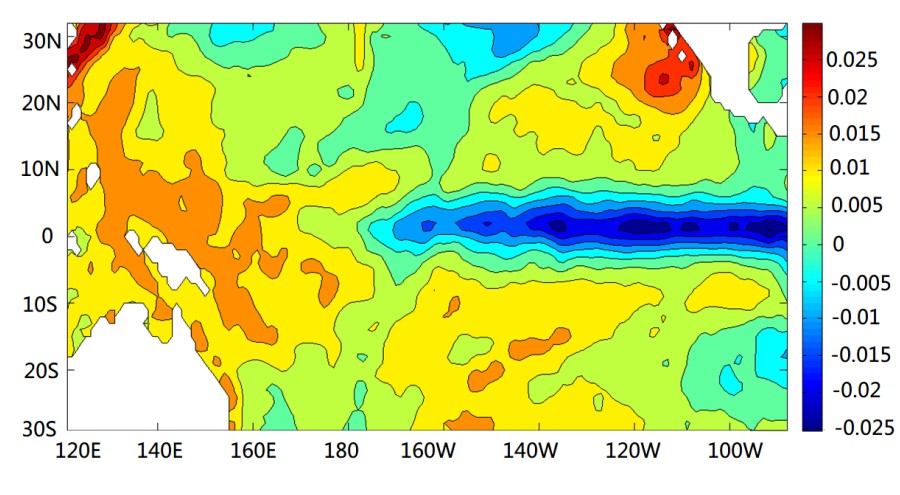


(c)


(d)

**Fig. 1** (a, c) First and second modes of the EOF deconstruction of the SSTA field, and (b, d) the
corresponding PC time series.



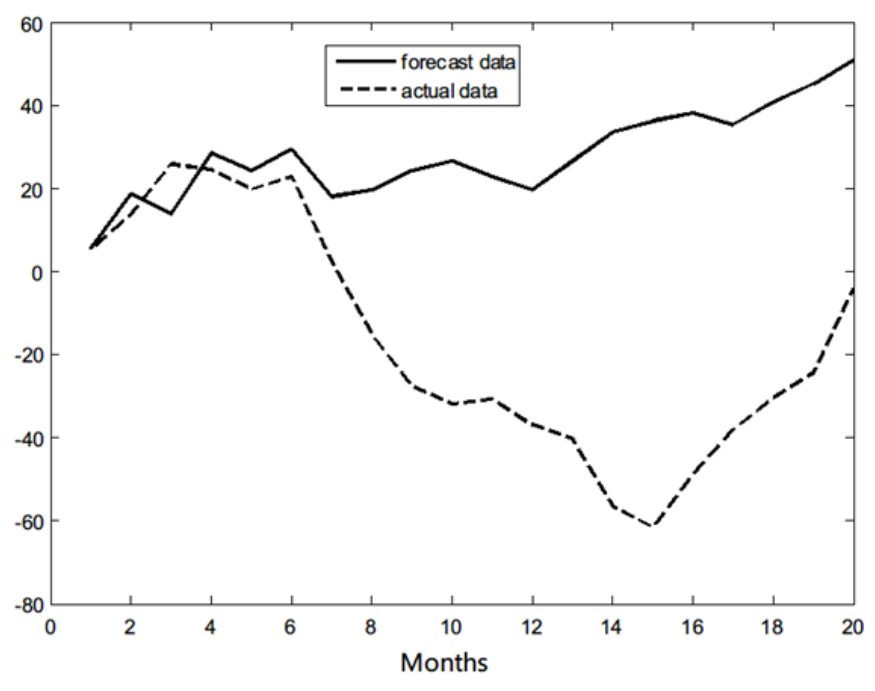


(a)

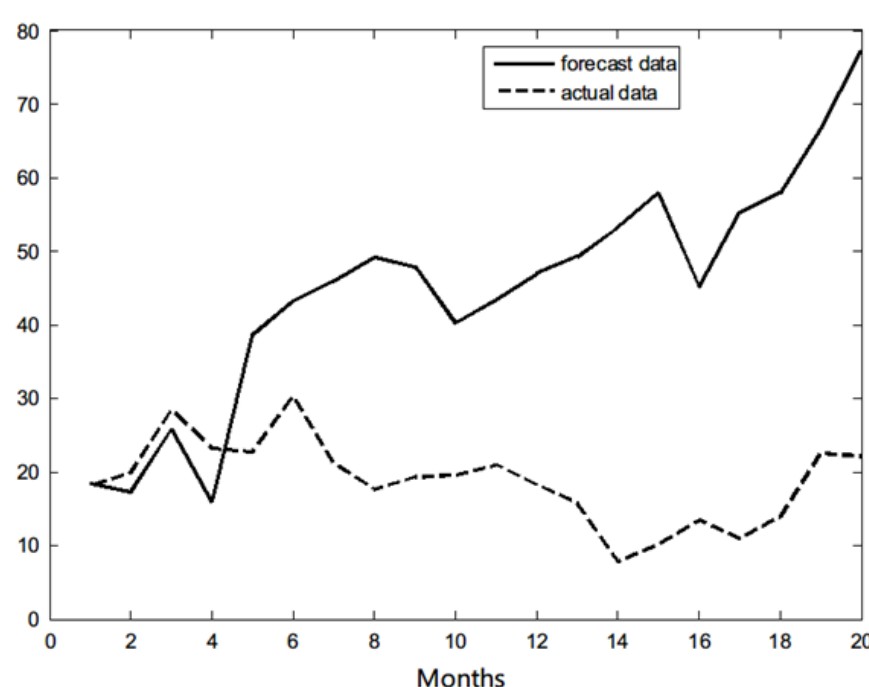




(b)

Fig.2 Forecast results of the first time coefficient series $T_1$    (a) and the second time coefficient series

$T_2$ (b)of the SSTA field by the original model














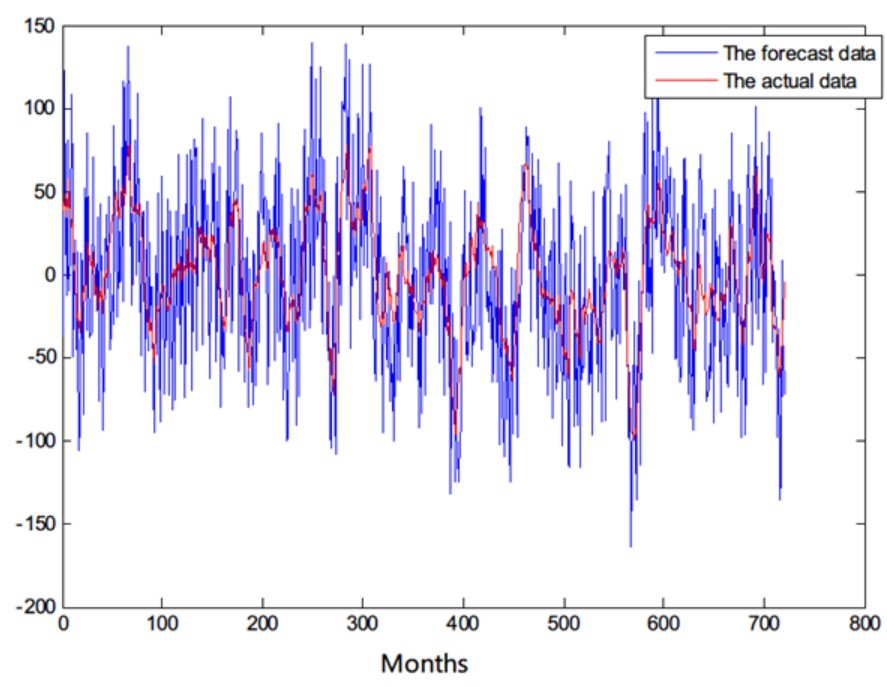


(a)

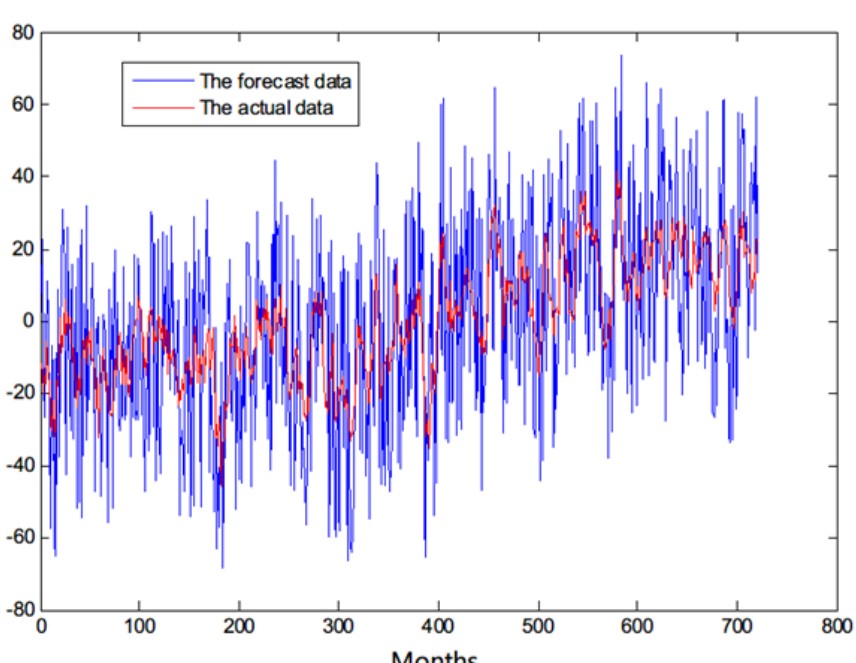






(b)

Fig.3The cross-validated retroactive hindcast results of the first time coefficient series $T_1$ (a)and the
second time coefficient series $T_2$ (b)of the SSTA field by the original model











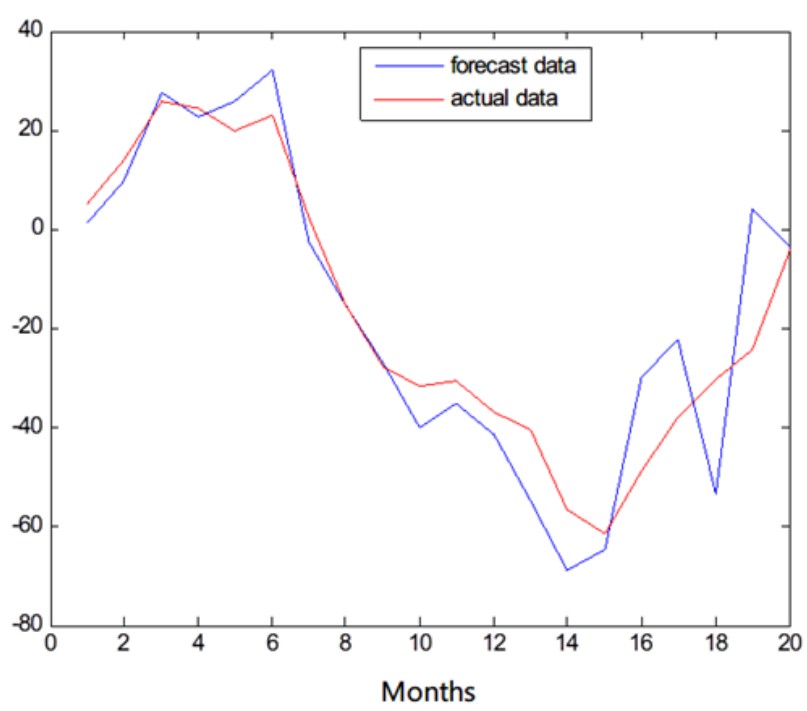


(a)





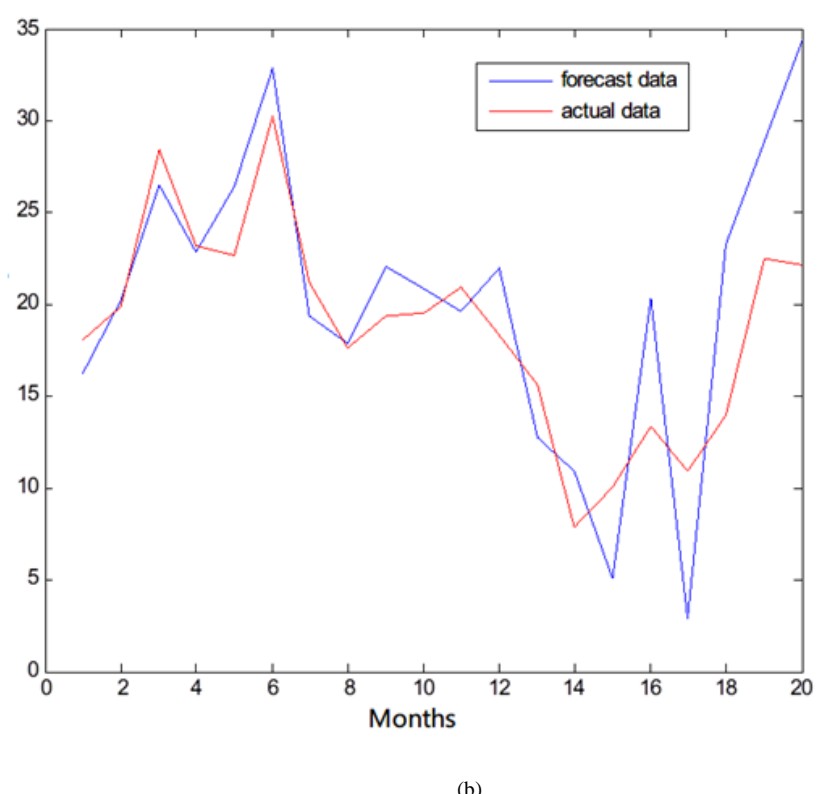


(b)

Fig. 4. Long-term step-by-step forecast results of the first time coefficient series $T_1$ (a)and the second
time coefficient series $T_2$ (b)of the SSTA field by the improved model


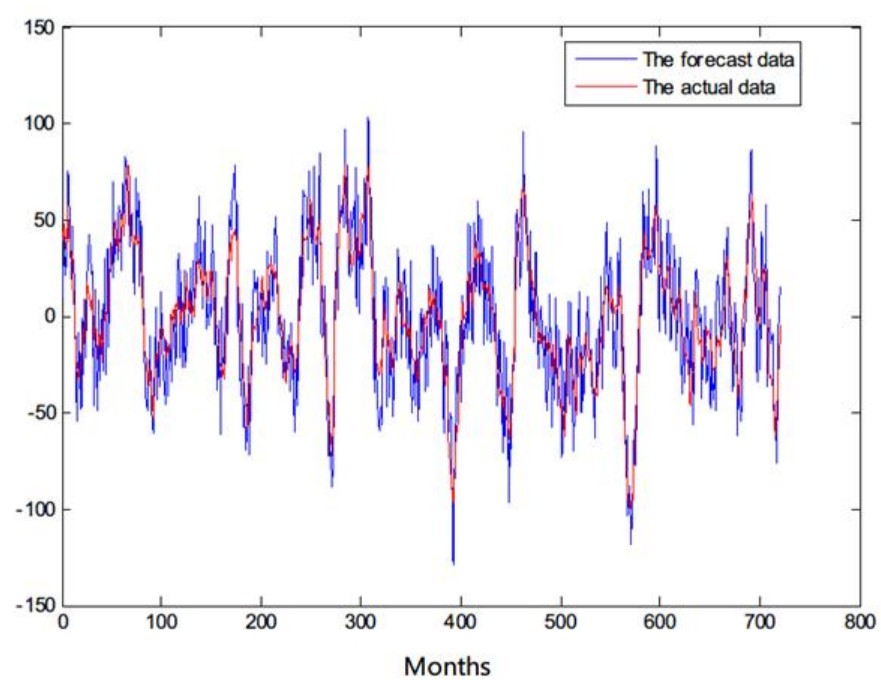


(a)

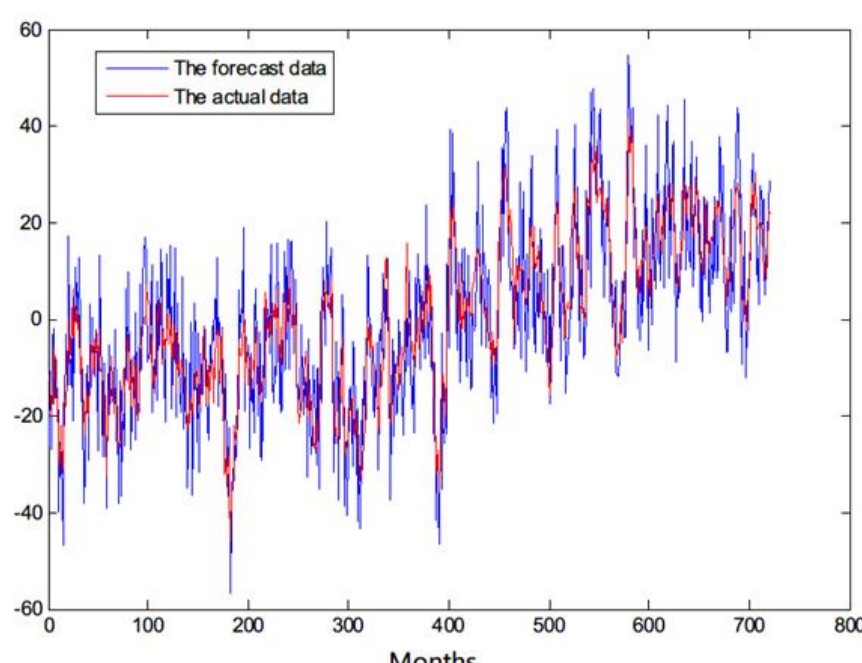


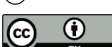



(b)

Fig. 5. The cross-validated retroactive hindcast results of the first time coefficient series $T_1$ (a)and the
second time coefficient series $T_2$ (b)of the SSTA field by the improved model










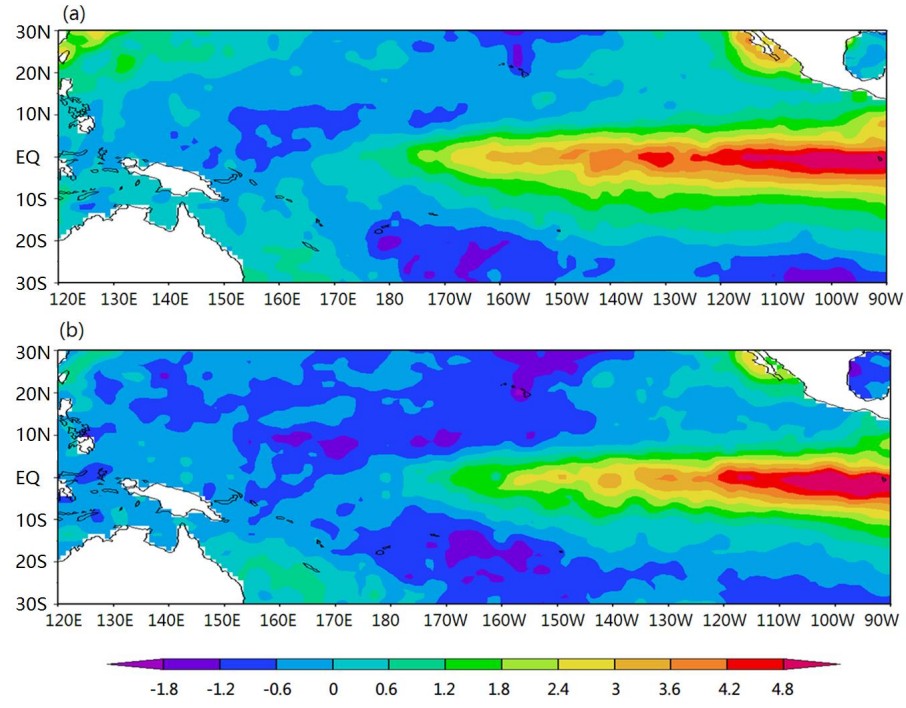


Fig.6. The forecast SSTA field(a) and the actual SSTA field (b)of an El Niño event (Dec.1997)














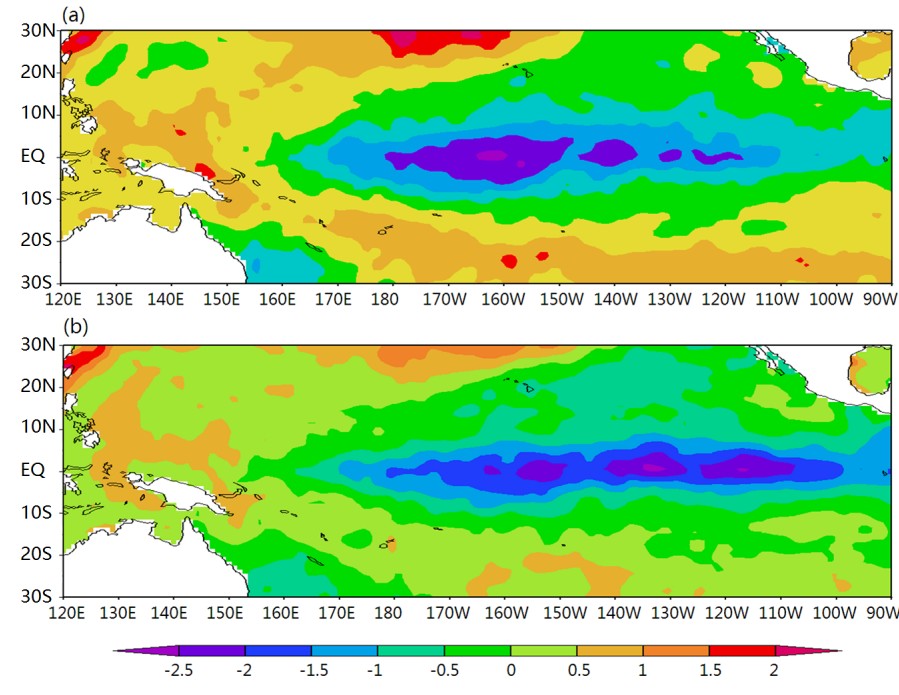


Fig.7. The forecast SSTA field(a) and the actual SSTA field (b)of a La Ni ña event (Dec.1999)













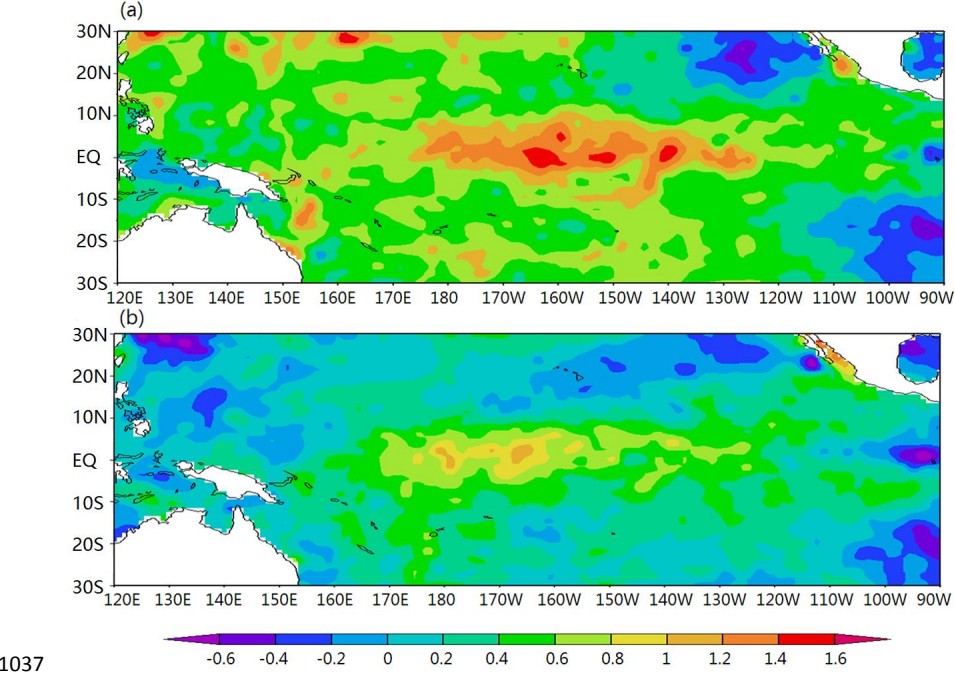


Fig.8. The forecast SSTA field(a) and the actual SSTA field (b)of neutral event (Nov.2002)





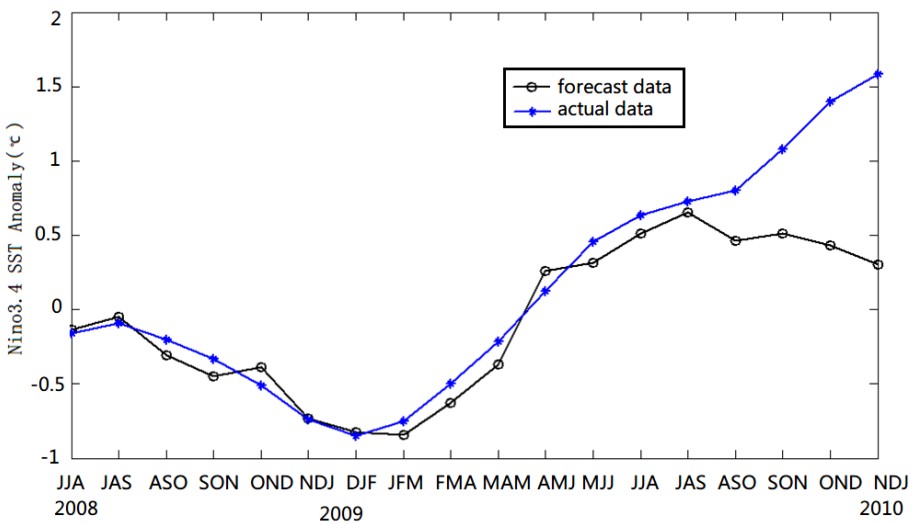


Fig.9. The improved dynamical-statistical model prediction of the ENSO index











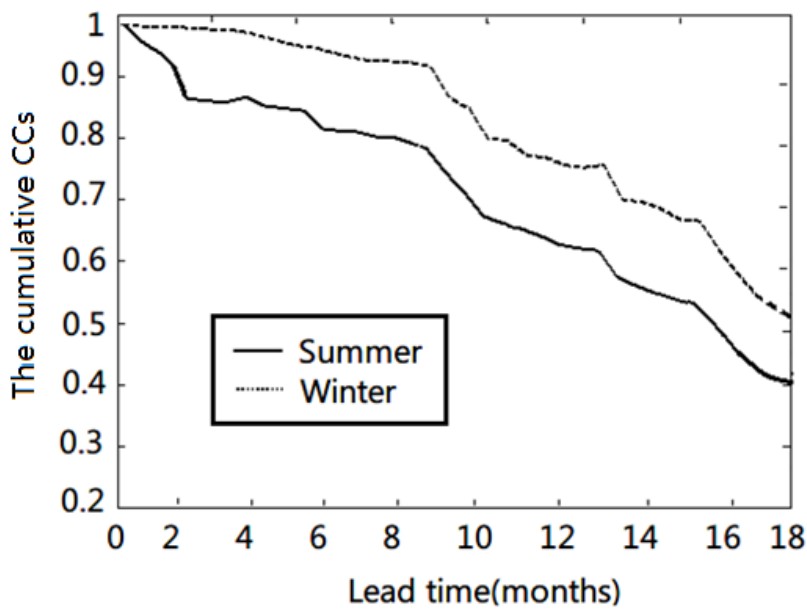


(a)

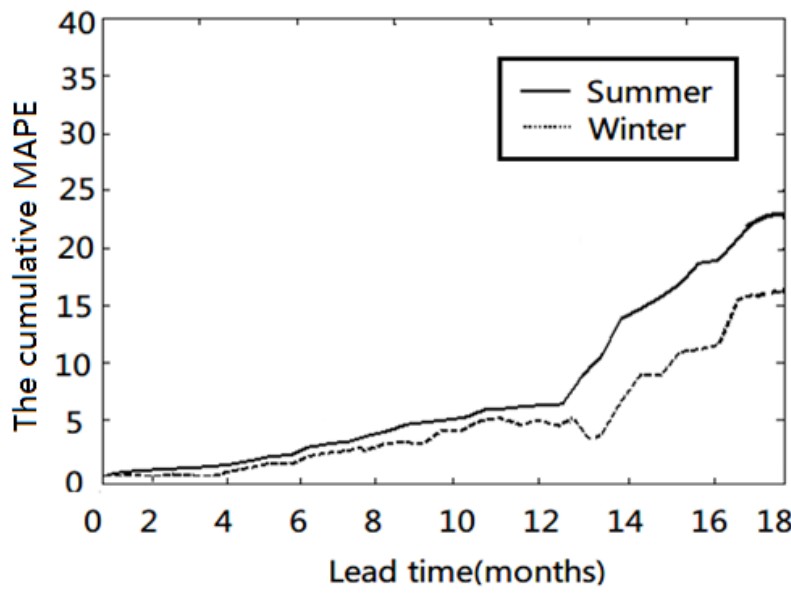


(b)
Fig.10. The cumulative correlation coefficients(CCs) (a) and cumulative mean absolute percentage
error(MAPE) (b) changing with time of different lead times






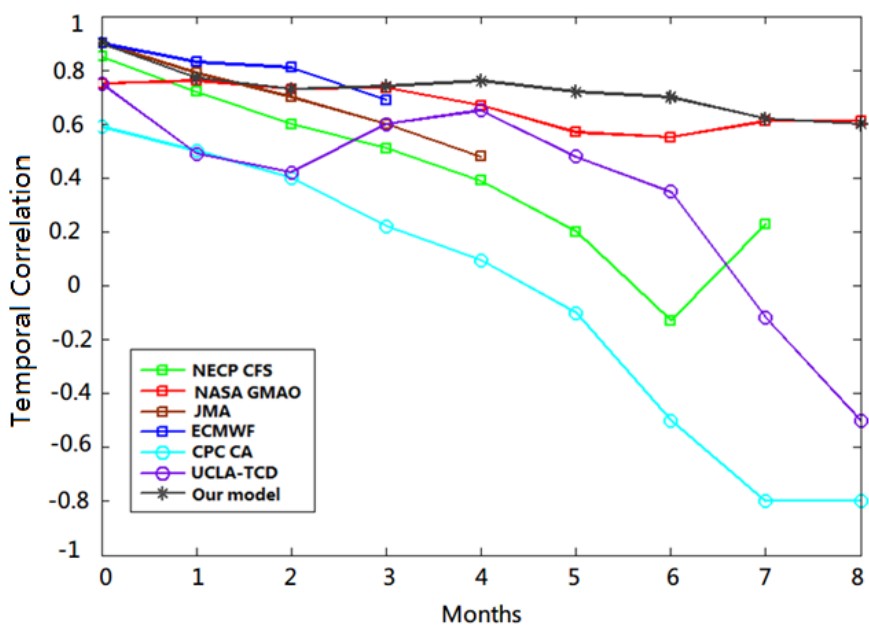


Fig. 11. Temporal correlation between model forecasts and observations for all seasons combined, as a
function of lead time. Each line highlights one model.


















Fig . 12. RMSE in standardized units, as a function of lead time for all seasons combined. Each line
highlights one model.


















**Table:**
**Table1.** The forecast results of the models of different variables

| The model | The forecast skill of 60 cross-validated retroactive hindcasts experiments of the ENSO index for all seasons combined at lead times of 8 months | |
|---|---|---|
| | the temporal correlation | the root mean square error |
| One variable($T_1$) | 0.5051 | 0.8075 |
| Two variables($T_1, T_2$) | 0.5613 | 0.7679 |
| Three variables($T_1, T_2, SOI$) | 0.6027 | 0.7275 |
| Four variables ($T_1, T_2, SOI, EAWMI$) | 0.6344 | 0.6728 |
| Five variables ($T_1, T_2, SOI, EAWMI, u_1$) | 0.5923 | 0.7344 |
| Six variables ($T_1, T_2, SOI, EAWMI, u_1, PNA$) | 0.5528 | 0.7806 |






**Table2.**The correlation coefficient(CC) and Mean absolute percentage error(MAPE) of long-term
fitting test when the retrospective order $p$ is different

| $p$ | | 4 | 5 | 6 | 7 | 8 | 9 | 10 |
|---|---|---|---|---|---|---|---|---|
| The forecast results of long-term fitting test | CC | 0.75 | 0.73 | 0.81 | 0.74 | 0.70 | 0.72 | 0.68 |
| | MAPE | 18.42% | 19.36% | 14.56% | 20.39% | 25.31% | 24.18% | 27.33% |
| $p$ | | 11 | 12 | 13 | 14 | 15 | 16 | |
| The forecast results of long-term fitting test | CC | 0.68 | 0.70 | 0.65 | 0.62 | 0.60 | 0.62 | |
| | MAPE | 28.10% | 26.58% | 30.91% | 33.14% | 34.97% | 33.56% | |

















**Table3.** The forecast results of $T_1$ and $T_2$ in different examples within 6 and 12 months

| Forecast events | The results within 6-months | | The results within 12-months | |
|---|---|---|---|---|
| | CC | MAPE | CC | MAPE |
| The average of 18 El Niño examples of $T_1$ | 0.824 | 8.45% | 0.719 | 12.67% |
| The average of 22 La Niña examples of $T_1$ | 0.846 | 7.68% | 0.740 | 11.28% |
| The average of 20 Neutral examples of $T_1$ | 0.885 | 6.23% | 0.789 | 9.85% |
| The average of total 60 examples of $T_1$ | 0.850 | 7.41% | 0.748 | 10.95% |
| The average of 18 El Niño examples of $T_2$ | 0.811 | 8.79% | 0.703 | 13.28% |
| The average of 22 La Niña examples of $T_2$ | 0.833 | 7.35% | 0.731 | 11.96% |
| The average of 20 Neutral examples of $T_2$ | 0.896 | 6.68% | 0.795 | 10.08% |
| The average of total 60 examples of $T_2$ | 0.842 | 7.64% | 0.740 | 11.71% |














**Table. 4**. Temporal correlation(CC) and the mean absolute percentage error (MAPE) between
model forecasts and observations within 12 months for Nov–Jan, Dec–Feb, and Jan–Mar as lead
time.of winter and for May-July, June-August and July-Sep. as lead time of summer.

| Forecast events | Lead time of all seasons combined | | Lead time of summer (MJJ-JJA-JAS) | | Lead time of winter (NDJ-DJF-JFM) | |
|---|---|---|---|---|---|---|
| | CC | MAPE | CC | MAPE | CC | MAPE |
| The average of 18 El Niño examples | 0.604 | 9.70% | 0.569 | 10.33% | 0.677 | 8.02% |
| The average of 22 La Niña examples | 0.625 | 8.97% | 0.581 | 9.82% | 0.695 | 7.83% |
| The average of 20 Neutral examples | 0.798 | 5.96% | 0.752 | 6.86% | 0.844 | 4.60% |
| The average of total 60 examples | 0.712 | 7.62% | 0.633 | 8.51% | 0.776 | 6.52% |















**Table5.** The correlation coefficients among four factors

| Correlation coefficients | $T_1$ | $T_2$ | SOI | EAWMI |
|---|---|---|---|---|
| $T_1$ | | 0.419 | 0.401 | 0.337 |
| $T_2$ | 0.419 | | 0.424 | 0.356 |
| SOI | 0.401 | 0.424 | | 0.408 |
| EAWMI | 0.337 | 0.356 | 0.408 | |

