# Peer review of "Forecasting experiments of a dynamical-statistical model"

_Ocean Science, 2017_

## Referee Comment (RC1) · Anonymous Referee #1 · 26 Nov 2017

Overview: Accurate prediction of ENSO event is crucial to improving climate prediction, however, it is often problematic in model, if even in national research center such as U.S. Climate Prediction Centre, Japan Meteorological Agency, and European Centre for Medium-Range Weather Forecasting, which still a challenge isuues. This paper introduced a new approach applied to SSTA field and ENSO index based on a dynamic system reconstruction idea and the principle of self-memorization. The overall results indicated that the improved model is more appropriate for describing both SSTA field and ENSO events. This study provides a useful information and a possible approach for the improving the ENSO prediction, especially there is potential in dynamical extent prediction, which can be accepted after carefully addressing the following comments.

[Figure]

Specific comments: 1. The method used in this study is based on the statistic regression, which basically depends on the quality of observations. In section 2.1, although the authors claimed that the monthly average SST data from the UK Met Office Hadley Centre is adopted in this study, the reliability of this datasets is not mentioned. Besides, the verification of this datasets with in-situ observation is also strongly recommended by this reviewer. 2. One important conclusion of this study is "The difference between forecast results in summer and those in winter is not high, indicating that the improved model can overcome the spring predictability barrier to some extent". This conclusion is vague and lack of rigorous verification because the authors did not verify their results in spring season. 3. Lines 42-44, "Compared with six mature models published previously, the present model has an advantage in prediction precision and length, and is a novel exploration of the ENSO forecast method". The major concerns of this reviewer are: what is the sample size in comparing the forecast results? Are those samples really representative?

Minor comments: 1. Line 122, give the full name of "SOI". 2. Line 549, "mode" should be "model".

---

## Referee Comment (RC2) · Anonymous Referee #2 · 14 Dec 2017

I have four major concerns with this paper 1. Construction of the first two predictors ie T1 and T2, 2. Selection of the other predictors, 3. Structure of the model, and 4. Model validation

Detail comments on the 4 points

1 . Section 2.2 EOF deconstruction. This section requires some more detail. While the given reference describes the EOF method, we need to know how it is applied here. Is the correlation or covariance matrix used? How are the anomalies constructed - simple removal of the monthly means? How are the anomalies smoothed - how strong is the smoothing and is it applied spatially or over time? More importantly, why are only the

first 2 EOFs considered? A similar analysis has recently been reported by L'Heureux et al (Clim Dyn 2013, DOI 10.1007/s00382-012-1331-2). Their first two EOFs are similar to those described here (but with no smoothing and hence lower explained variance). Using different data sets and time periods, they show that the 2nd EOF is not stable, being entirely due to the strong trend (also evident in Figure 1d). The pattern does not appear if the data is detrended, and also becomes less important if different time periods and/or domains are used. Most importantly, they do not interpret it as indicating "the ENSO signal beginning to decay".

2. Section 2.3 Predictor selection The selection of other potential predictors is confusing. Apart from T1 and T2, the other potential predictors come from a fairly limited set, and are not well supported by the referenced works. In lines 157-160, zonal winds in the western and eastern equatorial Pacific are mentioned, and it is well known that westerly wind anomalies in the western equatorial Pacific can (and do) trigger equatorially trapped oceanic Kelvin waves. There is an extensive amount of literature on the relationship between western equatorial Pacific zonal wind and ENSO, but here no references are given and only the eastern equatorial winds is considered. Trenberth et al discuss a link between ENSO and the PNA pattern (amongst other modes of extratropical variability), but this is the context of ENSO forcing of the PNA, ie ENSO leads to PNA teleconnections, but PNA does not predict ENSO. Yang et al introduce the EAWM index, but they note that "the relationship between ENSO and the east Asian winter monsoon is relatively weak". Nowhere do they suggest that the EAWMI is closely related to any ENSO indices. It is not surprising that the east Pacific wind and PNA do not feature in the final model

3. The model The remainder of section 2.3, concerned with determining the number of predictors is difficult to follow. It is not until section 3 (page11) that it is revealed that the model is a dynamical system of four second order coupled equations, involving the products of the various predictors as well as the predictors themselves. Nowhere is the inclusion of these terms discussed or justified. What physical processes do these

terms represent? What do the predictors squared represent?, and the cross products ie what do T1 * SOI or T2 * EAWMI mean? Since the model is not a linear regression model, is stepwise regression a valid procedure for determining the significance of the predictors?

line 195. The idea that a model with the number of predictors less than 10% of the sample size can avoid overfitting is new to me. The reference given (Tetko et al) is about neural networks. Is this applicable to the system of coupled equations used here? (I could only see the first page) Also I am not sure if the discussion in 198-203 is incorrect. Even if only 34 parametres are accepted, the full set of 56 parameters must be estimated to know which to accept or reject. This may be more a problem of introducing artificial skill, which has long been recognised as a problem in statistical forecasting. It generally arises when you try enough predictors, and retain those that "work" and discard the others.

This question of the number of parametrs / predictors is exacerabated in Section 4 and 5 where the number of predictors is increased again by including lagged values. On first inspection Equations 3 and 7 involve 112 parameters. There are 28 alphas, 28 thetas, as given in lines 395 and 396. (In line 202, it is stated that there are 28 self memorization parameters beta; but there are no betas in Eqs 3 and 5, but there are in Appendix B) In addition each of the four F "dynamical cores" involve 14 parameters as shown in Equation 1, assuming that the same F is used at each lagged time. Given that the input data (the xi) are different at each lag, is the same F a valid assumption? Even with the authors 34 accepted values in the Fs, there is still a total of 90 parameters. This is well over 10%, and on the authors own criterion, this would suggest that the system is perhaps overfit. Additionally, all the 720 observations are not statistically independent. Both T1 and the SOI (and probably T2 with its strong trend) are strongly auto-correlated, and the effective sample size is probably significantly less than 720. All in all, this discussion is very confusing!

4. Model Validation
line 281-288. This paragraph took me a long time to understand, especially how one could obtain correlations and MAPE values based on a single forecast. As I understand it, "at this time" refers to the forecast at five months, and the correlation and MAPE are calculated over the first five months forecasts, and in general the values at the Nth month are based on the first N months forecast. (I assume that this is the "n" in the equation for MAPE on line 283) This method would suggest that the correlation at one month is undefined, and 1.0 (perfectly accurate) at two months? This same type of calculation appears to be used in Tables 3 and 4. line 289-298. Another confusing paragraph. January 1951 to January 1952 inclusive? is 13, not 12 months. How was the omitted section forecast, ie was it simply a 12 (or 13) month forecast starting at the last point before the omitted data? it is difficult to judge how "good" the forecast was based on Figure 3. Again it is not clear how the correlation and MAPE statistics were calculated - only one value is given, so presumeably it is taken over all (720 months) forecast? However the discussion in lines 310-312 suggest that individual 12 month forecasts were also evaluated. Overall the discussion of the forecast process and its validation in not clear.

Some minor points (There are many minor points - these are just a few that stood out to me)

In line 170, all 4 data sets range from Jan 1951 to Jan 2010, yet in at least 4 places, lines 292, 373, 402 and 416 forecasts are evaluated up to December 2010?

lines 249-253. Why does normalising the raw values avoid the overfitting problem?.

line 254. What criterion is used to determine what are "weak items" with "small dimension coefficient"

line 280 "forecast performance ... was better" than what??

Section 6.2 - Table 5 The values reported here do not make sense. By construction, EOFs (the spatial patterns) are orthogonal, and the PCs (the time series) are uncorrelated. L'Heureax et al report that the correlation between PC1 and PC2 (using the same HADISST data set) is 0.4 when the time series are detrended. This is the same value quoted in Table 5. Has T2 been detrended here also?

EOF1 is the cannonical ENSO pattern, and its time series is stronly correlated with the standard Nino indices (l'Heureaux et al give a value of 0.94 between their first EOF and the Nino3.4 index). In turn the Nino3.4 index is strongly correlated to the SOI, so that is difficult to see the correlation beteen T1 and the SOI being as small as the 0.4 given in Table 5.(This correlation is where the term ENSO ie El Nino - Southern Oscillation arises)

Acronyms need to be defined the first time they are used, eg EOF on lines 128-130

Figure caption (line 912) for figure 1 in List of figures is incorrect, and different to that given with the figure itself (line 959).

References are incomplete; there are at least 15 references that are not cited in the text, and a number that are cited but referenced.

---

## Author Comment (AC1) · 6 Feb 2018

All the authors are extremely grateful to you for providing your excellent comments and valuable advices for this paper. Your major suggestions that the reliability of this datasets is not mentioned and the authors did not verify their results in spring season are very helpful for us. Based on your suggestions, we have made some revisions to on our paper. We have added the discussion of reliability of this datasets and the new results in spring season based on your specific comments. Thank you again for your valuable comments to improve our submission. If there are still any problems on the method, diction, phrasing, grammar, and spelling, please do not hesitate to

tell us and we'll try our best to improve them. The specific revision can be seen the Supplement files. There are three files in the *.zip: 1. Responses to reviewer#1.pdf 2.the manuscript with the remarked change 3. the clear revised manuscript. The specific revision can be seen in the three files.

Please also note the supplement to this comment:
https://www.ocean-sci-discuss.net/os-2017-78/os-2017-78-AC1-supplement.zip

---

## Author Comment (AC2) · 6 Feb 2018

All the authors are extremely grateful to you for providing your excellent comments and valuable advices for this paper. Your major four suggestions that Construction of the first two predictors ieT1 and T2; Selection of the other predictors; Structure of the model and Model validation are very helpful for us. Based on your suggestions, we have made major revisions to on our paper. We have added the discussion of the selection of the predictors, the structure of the model and the model validation based on your specific comments.

Thank you again for your valuable comments to improve our submission. If there are

still any problems on the method, diction, phrasing, grammar, and spelling, please do not hesitate to tell us and we'll try our best to improve them.

The specific revision can be seen the Supplement files. There are three files in the *.zip: 1. Responses to reviewer#2.pdf 2.the manuscript with the remarked change 3. the clear revised manuscript. The specific revision can be seen in the three files.

Please also note the supplement to this comment:
https://www.ocean-sci-discuss.net/os-2017-78/os-2017-78-AC2-supplement.zip

---

## Author Response (AR1)

**Dear Editor:**

Thank you very much for providing the opportunity for us to revise our paper.

Thank you very much for your contributions to this paper. And we are all extremely grateful for having a chance to make further improvements. Reading and considering all comments of two reviewers carefully, we have made major revisions on our paper. The major three suggestions of reviewer1 and detail comments on the 4 points of reviewer2 are very helpful for us. Following the two reviewers' suggestions, we have made major revisions on our paper.

Finally, we write the point-by-point response to answer the two reviewers' questions for better communication. If there are still any problems on the method, diction, phrasing, grammar, and spelling, please do not hesitate to tell us and we'll try our best to improve them.

Thank you again for your comments to improve our paper. Wish your journal better and better.

Yours,

Mei Hong

2018-02-06

**Responses to reviewer#1:**

All the authors are extremely grateful to you for providing your excellent comments and valuable advices for this paper. Your major suggestions that the reliability of this datasets is not mentioned and the authors did not verify their results in spring season are very helpful for us. Based on your suggestions, we have made some revisions to on our paper. We have added the discussion of reliability of this datasets and the new results in spring season based on your specific comments.

Thank you again for your valuable comments to improve our submission. If there are still any problems on the method, diction, phrasing, grammar, and spelling, please do not hesitate to tell us and we'll try our best to improve them.

In the following, kind comments you suggested before are in black text with corresponding actions taken by us following in blue.

Specific comments:

1. The method used in this study is based on the statistic regression, which basically depends on the quality of observations. In section 2.1, although the authors claimed that the monthly average SST data from the UK Met Office Hadley Centre is adopted in this study, the reliability of this datasets is not mentioned. Besides, the verification of this datasets with in-situ observation is also strongly recommended by this reviewer.

Responses:Good suggestions. In the previous paper, we have neglected the discussion of reliability of this datasets. Now there are three main categories of SST

data. The gridded $2°\times2°$ NOAA Extended Reconstructed SST dataset (ERSST.v3b;

Smith et al. 2008) includes in situ data (ships and buoys), but does not include satellite data. The gridded $1°\times1°$ Met Office Hadley Sea Ice and SST dataset (HadISST1; Rayner et al. 2003) includes both in situ and available satellite data. The gridded $1°\times1°$ NOAA Optimal Interpolation SST (OISST.v2; Reynolds et al. 2002)

incorporates in situ and satellite data, but unlike the other two SST datasets, it is only available in the recent period from November 1981 to the present. Both HadISST1

and ERSST.v3b are available from the mid-to-late 1800s, but only monthly data from

1951 to 2010 was considered in this study.

Considering comprehensively, the gridded $1°\times1°$ Met Office Hadley Sea Ice and

SST dataset data, no matter from data quality or data length, is the most appropriate to used.

The specific revision can be seen from line118 to line120 in page6.

We sincerely hope for your satisfaction with our revision. Thank you again for your kind suggestion.

References:

Smith TM, Reynolds RW, Peterson TC, Lawrimore J (2008) Improvements to

NOAA's historical merged land–ocean surface temperature analysis (1880–2006). J

Clim 21:2283–2296.

Rayner NA, Parker DE, Horton EB, Folland CK, Alexander LV, Rowell DP, Kent EC,

Kaplan A (2003) Global analyses of sea surface temperature, sea ice, and night marine air temperature since the late nineteenth century. J Geophys Res 108(D14):4407.

doi:10.1029/2002JD002670

Reynolds RW, Rayner NA, Smith TM, Stokes DC, Wang W (2002) An improved in situ and satellite SST analysis for climate. J Clim 15:1609–1625

2. One important conclusion of this study is "The difference between forecast results in summer and those in winter is not high, indicating that the improved model can overcome the spring predictability barrier to some extent". This conclusion is vague and lack of rigorous verification because the authors did not verify their results in spring season.

Responses:Good suggestions. The skill of forecasts that start in February or May drops faster than that of forecasts that start in August or November. This behavior, often termed the spring predictability barrier, is in part because predictions starting from February or May contain more events in the decaying phase of ENSO (Jin et al.,

2008). Based on the reviewer's suggestion, we have added the experiments in the spring and in the autumn in Table4. From the table, we can see the forecast result in spring of our model is also good, indicating that the improved model can overcome the spring predictability barrier to some extent. The specific revision can be seen in from page66.

We sincerely hope for your satisfaction with our revision. Thank you again for your kind suggestion.

**Table. 4**. Temporal correlation(TC) and the mean absolute percentage error (MAPE) between
model forecasts and observations within 12 months for Nov.–Jan., Dec.–Feb., and Jan.–Mar. as
lead time of winter, for Feb.–Apr. , Mar.–May and Apr.–June as lead time of spring, for May-July,
June-August and July-Sep. as lead time of summer and for August-Oct., Sep.-Nov. and Oct.-Dec.

as lead time of autumn.

| Forecast events | Lead time of all seasons combined | | Lead time of summer (MJJ-JJA-JAS) | | Lead time of autumn (ASO-SON-OND) | | Lead time of winter (NDJ-DJF-JFM) | | Lead time of spring (FMA-MAM-AMJ) | |
|---|---|---|---|---|---|---|---|---|---|---|
| | TC | MAPE | TC | MAPE | TC | MAPE | TC | MAPE | TC | MAPE |
| The average of 18 El Niño examples | 0.604 | 9.70% | 0.569 | 10.33% | 0.632 | 8.85% | 0.677 | 8.02% | 0.538 | 11.6% |
| The average of 22 La Niña examples | 0.625 | 8.97% | 0.581 | 9.82% | 0.645 | 8.41% | 0.695 | 7.83% | 0.579 | 9.82% |
| The average of 20 Neutral examples | 0.798 | 5.96% | 0.752 | 6.86% | 0.831 | 5.31% | 0.844 | 4.60% | 0.765 | 7.07% |
| The average of total 60 examples | 0.712 | 7.62% | 0.633 | 8.51% | 0.786 | 6.88% | 0.776 | 6.52% | 0.653 | 8.03% |

3. Lines 42-44, Compared with six mature models published previously, the present model has an advantage in prediction precision and length, and is a novel exploration of the ENSO forecast method". The major concerns of this reviewer are: what is the sample size in comparing the forecast results? Are those samples really representative?

Responses:Good suggestions. As shown in Table 4, our ENSO forecast is a total of 60

experiments, including 18 ElNino examples, 22 La Ni n a examples, and 20 Neutral examples, and each experiment contains lead time of four seasons. Finally, it is the equivalent of 240 experiments. Figure 11 and Figure 12 is the average TC and RMSE

of the 240 experiments of compared with six mature models, covers a variety of different types of ENSO and different lead time. So those samples should be really representative . We haven't explained it in previous paper, and now we explain it from line564 to 567 on page27.

We sincerely hope for your satisfaction with our revision. Thank you again for your kind suggestion.

Minor comments:

1.Line 122, give the full name of "SOI".

Responses:Good suggestions. Now we have given the full name of "SOI" as the

Southern Oscillation Index (SOI) in line128 in page6.

We sincerely hope for your satisfaction with our revision. Thank you again for your kind suggestion.

2. Line 549, "mode" should be "model".

Responses:Good suggestions. Now we have revised "mode" as"model" in line545 in page26.

We sincerely hope for your satisfaction with our revision. Thank you again for your kind suggestion.

# Responses to reviewer#2:

All the authors are extremely grateful to you for providing your excellent comments and valuable advices for this paper. Your major four suggestions that

Construction of the first two predictors ieT1 and T2; Selection of the other predictors;

Structure of the model and Model validation are very helpful for us. Based on your suggestions, we have made major revisions to on our paper. We have added the discussion of the selection of the predictors, the structure of the model and the model validation based on your specific comments.

Thank you again for your valuable comments to improve our submission. If there are still any problems on the method, diction, phrasing, grammar, and spelling, please do not hesitate to tell us and we'll try our best to improve them.

In the following, kind comments you suggested before are in black text with corresponding actions taken by us following in blue.

1 . Section 2.2 EOF deconstruction. This section requires some more detail. While the given reference describes the EOF method, we need to know how it is applied here. Is the correlation or covariance matrix used? How are the anomalies constructed

– simple removal of the monthly means? How are the anomalies smoothed - how strong is the smoothing and is it applied spatially or over time? More importantly, why are only the first 2 EOFs considered? A similar analysis has recently been reported by L'Heureux et al (Clim Dyn 2013, DOI 10.1007/s00382-012-1331-2).

Their first two EOFs are similar to those described here (but with no smoothing and hence lower explained variance). Using different data sets and time periods, they show that the 2nd EOF is not stable, being entirely due to the strong trend (also evident in Figure 1d). The pattern does not appear if the data is detrended, and also becomes less important if different time periods and/or domains are used. Most importantly, they do not interpret it as indicating "the ENSO signal beginning to decay".

Responses:Good suggestions. We have used covariance matrix,because the covariance matrix was selected to diagnose the primary patterns of co-variability in the basin-wide SSTs, rather than the patterns of normalized covariance (or correlation matrix). We have used the smooths function with MATLAB, which is five points two times moving, mainly filtering out some noise points and outliers.

Because the variance contribution of the first EOF mode is 61.33% and the variance contribution of the second EOF mode is 14.52%, so the first two EOF modes account for 75.85% of the total variance contribution, which has occupied most of the variance contribution and also contains most of the information of the field decomposition. So the first 2 EOFs are considered.

Based on the reference of L'Heureux et al. (Clim Dyn 2013, DOI

10.1007/s00382-012-1331-2), we need to do more experiments to prove that we choose the second mode of EOF to be appropriate, and whether different time periods will make us forecast unstable or not. Our original data is the monthly average SST

data from January 1951 to Dec. 2010, which are 60 years. We will increase the length of the data for 20 years (Jan.1931 –Dec.2010), for 10 years (Jan.1941- Dec.2010) and decrease the length of the data for 10 years (Jan.1961- Dec.2010), for 20 years (Jan.1971- Dec.2010). And then we use the same method to reconstruct a model and forecast the ENSO index as section5.4. The prediction results are shown in the following table:

     Table5. The forecast results of the different data periods

| Forecast events | The data periods (Jan. | The data periods (Jan. | The data periods (Jan. | The data periods (Jan. | The data periods(Jan. |
|---|---|---|---|---|---|

|  | 1951-Dec.2010) Lead time of all seasons combined | | 1931- Dec.2010) Lead time of all seasons combined | | 1941- Dec.2010) Lead time of all seasons combined | | 1961- Dec.2010) Lead time of all seasons combined | | 1971- Dec.2010) Lead time of all seasons combined | |
|---|---|---|---|---|---|---|---|---|---|---|
|  | TC | MAPE | TC | MAPE | TC | MAPE | TC | MAPE | TC | MAPE |
| The average of 18 El Niño examples | 0.604 | 9.70% | 0.683 | 9.02% | 0.642 | 9.35% | 0.572 | 10.15% | 0.551 | 10.44% |
| The average of 22 La Niña examples | 0.625 | 8.97% | 0.701 | 8.33% | 0.675 | 8.55% | 0.589 | 9.42% | 0.567 | 9.82% |
| The average of 20 Neutral examples | 0.798 | 5.96% | 0.845 | 5.12% | 0.821 | 5.56% | 0.746 | 6.21% | 0.721 | 6.58% |
| The average of total 60 examples | 0.712 | 7.62% | 0.771 | 7.14% | 0.740 | 7.38% | 0.680 | 7.96% | 0.652 | 8.15% |

From the table, we can see that in the 60 experiments, the prediction results of the data period increased by 20 years are the best, and the prediction results of the data period decreased by 20 years is the worst. This is because the more data we use, the more information it contain. But from the table we can also see the difference among forecast results of both TC and MAPE of five different sample data are less, and no abnormal change suddenly worse or better appear. All these indicate that using different data sets and time periods, even though may have a certain impact on the pattern of the 2nd EOF, but the impact on our forecast is not great and it will not make our forecast unstable.

The "indicating the ENSO signal beginning to decay" in our previous paper is a mistake of writing, which is not seen from the space mode of Figure 1 (c), but from the time mode of Figure 1 (d). From Figure1 (d) we can see the time coefficient has a significant upward trend over time, indicating "the ENSO signal beginning to enhanced".

We have added the discussion about the stability of our forecast in page6-7 and page28-29 and revised as "the ENSO signal beginning to enhanced " in page7.

We sincerely hope for your satisfaction with our revision. Thank you again for your kind suggestion.

2. Section 2.3 Predictor selection The selection of other potential predictors is confusing. Apart from T1 and T2, the other potential predictors come from a fairly limited set, and are not well supported by the referenced works. In lines 157-160, zonal winds in the western and eastern equatorial Pacific are mentioned, and it is well known that westerly wind anomalies in the western equatorial Pacific can (and do)

trigger equatorially trapped oceanic Kelvin waves. There is an extensive amount of literature on the relationship between western equatorial Pacific zonal wind and

ENSO, but here no references are given and only the eastern equatorial winds is considered. Trenberth et al. discuss a link between ENSO and the PNA pattern (amongst other modes of extratropical variability), but this is the context of ENSO

forcing of the PNA, ie ENSO leads to PNA teleconnections, but PNA does not predict

ENSO. Yang et al introduce the EAWM index, but they note that "the relationship between ENSO and the east Asian winter monsoon is relatively weak". Nowhere do they suggest that the EAWMI is closely related to any ENSO indices. It is not surprising that the east Pacific wind and PNA do not feature in the final model.

Responses:Good suggestions. Your opinion is very good. In pervious paper the factors that we may consider are relatively few. But we are a complex coupled model of four factor differential equations and are not the similar with a simple statistical model (such as stepwise regression). So in our pervious paper using the stepwise regression method to select factors also has a problem. According to your opinion, we have read more literatures. We have expanded the scope of factor selection and revised the criterion of selecting factors, and the paragraph has revised as follows:

Considering the complexity of computation, the amount of variables in the equations of our model can't be too large, usually 3 or 4 for the best. This has been explained in our previous studies (Zhang et al., 2006; Zhang et al., 2008). If there are more than 4 variables in the modeling equation, it will cause the amount of parameters such as $a_1, a_2, \ldots a_n, b_1, b_2, \ldots b_n, \ldots$ too large. The huge computation makes it difficult to be precisely modeled. Thus, the total number of parameters in the model of five variables was 102, which may cause an overfitting problem. Hence, when we selected the model of five or six variables which entailed large amounts of computation that made precision difficult, and too many parameters might cause an overfitting phenomenon. If we choose only two or even fewer variables, the forecast performance is poor too. Too few variables cause too small reconstructed parameters, resulting in amounts of important information missing out in the model. Thus, four variables are best for dynamically and accurately modeling. Because we have chosen two time series in section2.2 as the modeling objects, now we should select the other two ENSO intensity impact factors.

The ENSO intensity impact factor is an important issue in ENSO prediction.

Previous studies have been completed in this area, which found that teleconnection patterns, temperature, precipitation, wind and SSH may affect ENSO strength. For example, Trenberth et al. (1998) noted that PNA, SOI and OLR in the Pacific

Intertropical Convergence Zone (ITCZ) are all closely related to ENSO.

Webster(1999) pointed out after the 1970, Indian Ocean dipole (IOD) is not only affected by ENSO, but also affected the strength of ENSO (Ashok et al., 2001). Yoon and Yeh (2010) reported that the Pacific Decadal Oscillation (PDO) disrupts the linkage between El Ni˜no and the following Northeast Asian summer monsoon (NEASM) through inducing the Eurasian pattern in the mid-high latitudes. The vast majority of studies (Tomita and Yasunari, 1996; Zhou and Wu, 2010; Kim et al.,

2017)have concentrated on the impacts of ENSO on the East Asian winter monsoon( EAWM). During the EAWM season, ENSO generally reaches its mature phase and has the most prominent impact on the climate. Wang et al. (1999a) and

Wang et al. (1999b) suggested that the zonal wind factors in the eastern and western equatorial Pacific play a critical role in the phase of transition of the ENSO cycle, which could excite eastward propagating Kelvin waves and affect the SSTA in the equatorial Pacific. Zhao et al. (2012) analyzed the characteristics of the tropical

Pacific SSH field and its impact on ENSO events.

Based on the above analysis, we have selected nine factors, which may be closely related with the ENSO index (Niño3.4).

(1)The zonal wind in the eastern equatorial Pacific factor (u1) was calculated as the grid-point average of zonal wind in the area [5 °S ~ 5 °N, 150 °W ~ 90 °W].

(2) The zonal wind in the western equatorial Pacific factor (u2) was calculated as the grid-point average of zonal wind in the area [0 °~ 10 °N; 135 °E ~ 180 °E].

(3) The PNA teleconnection factor was obtained from the CPC.

(4) the dipole mode index factor (DMI) was obtained from SSTA for June-July-August (JJA) based on Saji(1999) method.

(5) The SOI factor was obtained from the CPC.

(6) The PDOI factor was obtained from department of Atmospheric Sciences in the university of Washington. The web is http://tao.atmos.washinton.edu/pdo/RDO.latest.

(7) The EAWM index (EAWMI) factor was proposed by Yang et al. (2002), which is defined by the meridional 850-hPa winds averaged over the region (20 ° ~40 °N, 100 °~140 °E).

(8) The OLR in the ITCZ factor was calculated as the grid-point average of OLR in the area [10 °N～20 °N,120 °E～150 °E].

(9) The SSH factor was calculated as the grid-point average of the SSH data in the area [10 °S ~ 10 °N; 120 °E ~ 60 °W].

A correlation analysis of the above factors was carried out and the results are shown in Table 2.

Table 2 shows that SOI and EAWMI have the stronger correlation with the front two time series $T_1, T_2$ than the other 7 factors. The results are also consistent with previous research (Clarke and Van Gorder, 2003; Drosdowsky, 2006; Zhang et al.,

1996; Wang et al., 2008; Yang and Lu, 2014). Therefore, the first time series $T_1$, the second time series $T_2$, SOI and EAWMI will be selected as prediction model factors.

Table 2. The correlation analysis between the front two time series $T_1, T_2$ and nine impact factors

| factors | $u_1$ | $u_2$ | PNA | DMI | SOI | PDOI | EAWMI | OLR | SSH |
|---------|-------|-------|------|--------|--------|--------|--------|--------|--------|
| $T_1$ | 0.3161 | 0.5684 | 0.4386 | -0.3457 | 0.7734 | 0.4081 | 0.6284 | 0.3287 | 0.3363 |
| $T_2$ | 0.2118 | 0.4181 | 0.2560 | -0.2345 | 0.5232 | 0.3065 | 0.4825 | 0.1816 | 0.2169 |

Actually, how many variables and which variables are used in our model become a key issue to be resolved. We are a complex four factor differential equations coupling model. We are a complex coupled model of four factor differential equations, so we are more concerned with the correlation between each other. The correlation must be considered as an important criterion to select the factors, but in order to further verify the correctness of the selection criterion, we have carried out the prediction experiments (the 60 cross-validated retroactive hindcasts experiments of the ENSO index for all seasons combined at lead times of 8 months ) of different variables. The forecast results of the models of different variables are as following:

Table3.The forecast results (The temporal correlation (TC) and the root mean square error (RMSE) )of the models of different variables

| The forecast | Three variables of the model | | |
|--------------|------------------------------|--|--|

| results | | | | | | |
|---|---|---|---|---|---|---|
| | $T_1,T_2,u_1$ | $T_1,T_2,u_2$ | $T_1,T_2,$PNA | $T_1,T_2,$DMI | $T_1,T_2,$SOI | $T_1,T_2,$PDOI |
| TC | 0.4423 | 0.5628 | 0.3852 | 0.3226 | **0.6027** | 0.3809 |
| RMSE | 0.9025 | 0.7855 | 0.9244 | 1.0041 | **0.7275** | 1.0642 |
| | $T_1,T_2,$EAWMI | $T_1,T_2,$OLR | $T_1,T_2,$SSH | | | |
| TC | 0.5829 | 0.3205 | 0.4288 | | | |
| RMSE | 0.7516 | 0.9814 | 0.9090 | | | |
| | four variables of the model | | | | | |
| | $T_1,T_2,u_1,u_2$ | $T_1,T_2,u_1,$PNA | $T_1,T_2,u_1,$DMI | $T_1,T_2,u_1,$SOI | $T_1,T_2,u_1,$ PDOI | $T_1,T_2,u_1,$ EAWMI |
| TC | 0.4672 | 0.3628 | 0.5617 | 0.5201 | 0.5028 | 0.5822 |
| RMSE | 0.8824 | 0.9902 | 0.7617 | 0.8233 | 0.8092 | 0.7132 |
| | $T_1,T_2,u_1,$OLR | $T_1,T_2,u_1,$SSH | $T_1,T_2,u_2,$PNA | $T_1,T_2,u_2,$DMI | $T_1,T_2,u_2,$SOI | $T_1,T_2,u_2,$ PDOI |
| TC | 0.3815 | 0.4128 | 0.3107 | 0.4125 | 0.5910 | 0.5504 |
| RMSE | 0.9702 | 0.9017 | 1.0255 | 0.9392 | 0.7128 | 0.7503 |
| | $T_1,T_2,u_2,$ EAWMI | $T_1,T_2,u_2,$OLR | $T_1,T_2,u_2,$SSH | $T_1,T_2,$PNA,DMI | $T_1,T_2,$PNA,SOI | $T_1,T_2,$PNA, PDOI |
| TC | 0.6048 | 0.4528 | 0.5308 | 0.3022 | 0.3875 | 0.2876 |
| RMSE | 0.6910 | 0.9028 | 0.8344 | 1.0578 | 0.9706 | 1.1305 |
| | $T_1,T_2,$PNA, EAWMI | $T_1,T_2,$PNA, OLR | $T_1,T_2,$PNA, SSH | $T_1,T_2,$DMI, SOI | $T_1,T_2,$DMI, PDOI | $T_1,T_2,$DMI, EAWMI |
| TC | 0.3527 | 0.2556 | 0.2175 | 0.5688 | 0.2028 | 0.5807 |
| RMSE | 0.9518 | 1.2024 | 1.3244 | 0.7425 | 1.2905 | 0.7015 |
| | $T_1,T_2,$DMI, OLR | $T_1,T_2,$DMI, SSH | $T_1,T_2,$SOI, PDOI | $T_1,T_2,$SOI, EAWMI | $T_1,T_2,$SOI,OLR | $T_1,T_2,$SOI, SSH |
| TC | 0.3504 | 0.4833 | 0.6022 | **0.6344** | 0.5876 | 0.5476 |

| RMSE | 1.1624 | 0.8530 | 0.7054 | **0.6728** | 0.7408 | 0.7895 |
|---|---|---|---|---|---|---|
| | $T_1,T_2,\text{PDOI},$ EAWMI | $T_1,T_2,\text{PDOI},$ OLR | $T_1,T_2,\text{PDOI},$ SSH | $T_1,T_2,\text{EAWMI},$ OLR | $T_1,T_2,\text{EAWMI},$ SSH | $T_1,T_2,\text{OLR},$ SSH |
| TC | 0.4217 | 0.2017 | 0.2044 | 0.5872 | 0.4607 | 0.2028 |
| RMSE | 0.9147 | 1.2085 | 1.2542 | 0.7233 | 0.8925 | 1.3524 |

From the table, we can see that for all the forecast results of the models of different variables, the prediction results of $T_1,T_2,\text{SOI}$ is the best among those of the three factors and the prediction result of $T_1,T_2,\text{SOI},\text{EAWMI}$ is the best among those of the four factors. But the prediction result of $T_1,T_2,\text{SOI},\text{EAWMI}$ is best among all, which proves that our selection factors are correct. In our previous study (Hong et al., 2015), the model of the Western Pacific subtropical high was established by using the correlations as a criterion to select factors and their forecast results are also good. Now we use the correlations as a criterion to select factors is also in line with our previous research.

With the deepening of the research, there are still a lot of new literatures that reveal the relationship between ENSO and the East Asian winter monsoon. For example:

[1] Kim Ji-Won ,Soon-Il An,Sang-Yoon Jun,Hey-Jin Park,Sang-Wook Yeh. 2017.ENSO and East Asian winter monsoon relationship modulation associated with the anomalous northwest Pacific anticyclone, Climate Dynamics, Volume 49, Issue 4, pp 1157–1179.

[2] Yang Se-Hwan and Lu Riyu . 2014. Predictability of the East Asian winter monsoon indices by the coupled models of ENSEMBLES, Advances in Atmospheric Sciences, Volume 31, Issue 6, pp 1279–1292.

[3] Wang L., Chen W. and Huang R. H., 2008. Interdecadal modulation of PDO on the impact of

ENSO on the east Asian winter monsoon, Geophysical Research Letter, DOI:

10.1029/2008GL035287.

So there is a good correlation between ENSO and the East Asian winter monsoon.

The specific revision can be seen in section2.3 in page7-10 and line616 to632 in page29-30.We sincerely hope for your satisfaction with our revision. Thank you again for your kind suggestion.

3-1. The remainder of section 2.3, concerned with determining the number of predictors is difficult to follow. It is not until section 3 (page11) that it is revealed that the model is a dynamical system of four second order coupled equations, involving the products of the various predictors as well as the predictors themselves. Nowhere is the inclusion of these terms discussed or justified. What physical processes do these terms represent? What do the predictors squared represent?, and the cross products ie what do T1 * SOI or T2 * EAWMI mean? Since the model is not a linear regression model, is stepwise regression a valid procedure for determining the significance of the predictors?

Responses:Good suggestions. Your opinion is very good. Based on your suggestion of question2, we have revised the discussion of how to determine the number of predictors. Our model is not a linear regression model, the stepwise regression may be a valid procedure for determining the significance of the predictors, so we also have revised the method for determining the significance of the predictors, the specific revision can be seen our answer of the question2.

The inclusion of these terms and the physical processes do these terms represent are important, especially for the discussion of dynamical characteristics of the dynamical model. But now we are difficult to give a clear meaning. Now the main work of our paper is the prediction experiments of the model. For the reason of time and length, this paper mainly discusses the prediction results of the model. The physical processes do these terms represent and the discussion of the dynamical characteristics of the model will be the focus of our next work. Before this, we have also used the Takens' delay embedding theorem to reconstruct the dynamical model of the Western Pacific subtropical high(WPSH). And Based on the reconstructed dynamical model, dynamical characteristics of WPSH are analyzed and an aberrance mechanism is developed, in which the external forcings resulting in the WPSH

anomalies are explored, which have been published(Hong et al., 2016). We also study the bifurcation and catastrophe of the West Pacific subtropical high ridge index of a nonlinear model (Hong et al., 2017). Based on our previous method and work, our next work is to analyze the physical processes and the dynamical characteristics of the

SST field.

The specific revision can be seen from line689 to704 in page33.We sincerely hope for your satisfaction with our revision. Thank you again for your kind suggestion.

(I could only see the first page) Also I am not sure if the discussion in 198-203 is incorrect. Even if only 34 parametres are accepted, the full set of 56 parameters must be estimated to know which to accept or reject. This may be more a problem of introducing artificial skill, which has long been recognised as a problem in statistical forecasting. It generally arises when you try enough predictors, and retain those that

"work" and discard the others.

This question of the number of parametrs / predictors is exacerabated in Section 4 and where the number of predictors is increased again by including lagged values. On first inspection Equations 3 and 7 involve 112 parameters. There are 28 alphas, 28

thetas, as given in lines 395 and 396. (In line 202, it is stated that there are 28 self memorization parameters beta; but there are no betas in Eqs 3 and 5, but there are in

Appendix B) In addition each of the four F "dynamical cores" involve 14 parameters as shown in Equation 1, assuming that the same F is used at each lagged time. Given that the input data (the xi) are different at each lag, is the same F a valid assumption?

Even with the authors 34 accepted values in the Fs, there is still a total of 90

parameters. This is well over 10%, and on the authors own criterion, this would suggest that the system is perhaps overfit. Additionally, all the 720 observations are not statistically independent. Both T1 and the SOI (and probably T2 with its strong trend) are strongly auto-correlated, and the effective sample size is probably significantly less than 720. All in all, this discussion is very confusing!

Responses:Good suggestions. Our final number of 90 parameters is still a little large for a sample size of 720. In the previous paper, this discussion of overfitting is a little confusing. So it is still necessary to further discuss whether our model has the overfitting problem or not. Thank reviewers to remind us this problem.

The definition of overfitting: The learned hypothesis may fit the training set very well, but fail to predict to new examples (fail to fit additional data or predict future observations reliably).

The potential for overfitting depends not only on the number of parameters and data but also the conformability of the model structure with the data shape, and the magnitude of model error compared to the expected level of noise or error in the data(Burnham and Anderson, 2002). So there are many reasons causing the overfitting phenomenon. But this does not mean having many parameters relative to the number of observations inevitably causes the overfitting problem (Golbraikh et al., 2003).

There is no evidence that more parameters will be certain to result in overfitting.

Based on the definition of overfitting and the previous studies(Golbraikh et al., 2003;

Everitt and Skrondal,2010), we can judge whether a model is overfitting or not by the accuracy of prediction results of independent samples (Golbraikh and Tropsha, 2002;

Qi and Li, 2006).

In the sample training, our model does not purposely pursue the high degree of the training samples fitting and improve the effectiveness of the independent generalization. In fact in our paper the forecast results of the Cross-validated retroactive hindcasts (section 5.2) and the independent samples validation (table3 and table4) are both good. Especially, the independent samples validation of the ENSO

index as the table4, we have carried out the 240 independent sample validation prediction of four seasons of different ENSO events and the coverage of independent samples test is very wide. Moreover, compared with 6 mature prediction models, the forecast results of our model are also good, which prove the overfitting problem does not exist in our model. According to the previous literature (Islam and Sivakumar,

2002; Sivakumar et al.,2001), we can see that prediction principle and structure of the phase space reconstruction (PSR) of dynamical system is not the same with the traditional neural network and in the small sample situation the forecasting results of

PSR model are better than those of the traditional neural network (Sivakumar et al. ,2002), which can be verified in the independent sample test (table3 and table4). So according to the definition of overfitting, we can say the over fitting phenomenon does not exist in our model.

Now we have added the new discussion of the overfitting problem from line633

to663in page30-31.

We sincerely hope for your satisfaction with our revision. Thank you again for your kind suggestion.

4-1.line 281-288. This paragraph took me a long time to understand, especially how one could obtain correlations and MAPE values based on a single forecast. As I

understand it, "at this time" refers to the forecast at five months, and the correlation and MAPE are calculated over the first five months forecasts, and in general the values at the Nth month are based on the first N months forecast. (I assume that this is the "n" in the equation for MAPE on line 283)

Responses:Good suggestions. Your understanding is right. "at this time" refers to the forecast at five months, and the correlation and MAPE are calculated over the first five months forecasts, and in general the values at the Nth month are based on the first

N months forecast. Now we revise the sentence "Using $T_1$ as an example, at this time, the temporal correlation between model predictions and corresponding observations was 0.8966 and the mean absolute percentage error (MAPE) (Hu et al.,

2001), $\mathrm{MAPE} = \dfrac{1}{n}\sum\limits_{i=1}^{n}\left|\dfrac{D_e(i) - D_0(i)}{D_0(i)}\right| \times 100$, was 8.32%." as "Using     as an example, the CC between model predictions and corresponding observations over the first five months forecasts was 0.8966 and MAPE was 8.32%. " for readers' better understanding.

The specific revision can be seen from line275 to276 in page13. We sincerely hope for your satisfaction with our revision. Thank you again for your kind suggestion.

4-2. This method would suggest that the correlation at one month is undefined, and

1.0 (perfectly accurate) at two months? This same type of calculation appears to be used in Tables 3 and 4.

Responses:Good suggestions. In previous paper, we have not explained the concept of correlation. There two different correlations in our paper. The first correlation in our paper is the pearson correlation coefficient (CC), which also can be called the linear correlation coefficient. It measures the strength and direction of a linear relationship between two variables (for example model output and observed values).

The mathematical formula for computing r is:

$$r = \frac{\sum_{i=1}^{n}(D_e(i) - \bar{D}_e) \cdot (D_0(i) - \bar{D}_0)}{\sqrt{\sum_{i=1}^{n}(D_e(i) - \bar{D}_e)^2 \cdot \sum_{i=1}^{n}(D_0(i) - \bar{D}_0)^2}}$$

Where n is the number of pairs of data, $D_e, D_0$ is a series of n observations and n
forecast values.
The CC (Wang et al. 2009) and the mean absolute percentage error (MAPE)( Hu et al. 2001) are employed as objective functions to calibrate the model. The CC

evaluates the linear relationship between the observed and predicting values and

MAPE measures the difference between the observed and predicting values. The forecast results of $T_1, T_2$ in Section3, table2 and table3 have used the above two evaluation criteria (r and MAPE).

While the evaluation criteria of the ENSO index in table4 is the temporal correlation (TC), its definition and specific calculation steps can be seen in these literatures (Kathrin et al.,2016; Nicosia et al. 2013); The TC is often used to measure the prediction effect of the ENSO index. For example, in 1995,Chen et al. used TC as the evaluation criteria to test the improved Predictability of El Nino Forecasting of their model and Barnston et al.in 2012 also used the TC to compare the forecast skill of 21 real-time seasonal ENSO models.

In the previous paper, we didn't explain two different correlations clearly, which will be easy for readers to misunderstand. Now we have explained two different correlations and the specific revision can be seen in all my paper.

We sincerely hope for your satisfaction with our revision. Thank you again for your kind suggestion.

Responses:Good suggestions. From Fig3, the prediction values (blue line) and the actual values (red line) are relatively close in some places, but in many places, especially in the peaks, the error is large, which in accordance with the analysis of Figure 2. The forecast results within 5 months of the simple dynamical reconstruction model in section3 are good, but the long term prediction results after 5 months become bad and the error increases quickly. So this is why we have to introduce the self -memorization principle to improve the long term prediction results.

We sincerely hope for your satisfaction with our revision. Thank you again for your kind suggestion.

4-5.Again it is not clear how the correlation and MAPE statistics were calculated - only one value is given, so presumeably it is taken over all (720 months) forecast?

Responses:Good suggestions. In pervious paper we haven't explained clearly how the correlation and MAPE statistics in Fig.3 were calculated. It isn't taken over all (720 months) forecast when only one value is given (The forecast for such a long time is not possible). The figure 3 merges the 60 experiments (each experiment is the prediction of the 12 month similar as Fig.2) on one picture. The Fig.3 is equivalent to experiments instead of the results of only one experiment, because the results of one experiment are not entirely representative. And through multiple cross experiments can more objectively reflect the forecast capability of our model. So the forecast results of 60 cross experiment (each experiment is the prediction of the 12

month as Fig.2) according to the time sequence can merger into a new time series (from Jan.1951-Dec.2010), and then the pearson correlation coefficient (CC) and the mean absolute percentage error (MAPE) can be calculated by the new prediction time series and the time series of the actual value based on the formula in the above answer of 4-2 problems. Actually, the CC and MAPE are the average of the prediction values of the 60 cross experiments. That's how the correlation and MAPE statistics were calculated in Fig. 3.

Now we have added the above explanation from line 294 to 300 in page14 for readers' better understanding.

We sincerely hope for your satisfaction with our revision. Thank you again for your kind suggestion.

4-6. However the discussion in lines 310-312 suggest that individual 12 month forecasts were also evaluated. Overall the discussion of the forecast process and its validation in not clear.

Responses:Good suggestions. The CC and MAPE in Fig.3 are the average of the prediction values of the 60 cross experiments. But each MAPE value of the above 60

experiments is not the same and the difference between the maximum and the minimum MAPE value is quite large, which means that the prediction results of the simple dynamical reconstruction model in section3 is not stable. So that is another reason why we need to introduce self -memorization principle to improve our model.

We sincerely hope for your satisfaction with our revision. Thank you again for your kind suggestion.

Some minor points

1.      In line 170, all 4 data sets range from Jan 1951 to Jan 2010, yet in at least 4 places,

Responses:Good suggestions. Now we have deleted the other 3 places about the description of the length of the data. And in pervious paper, " all 4 data sets from Jan.

to Jan. 2010" is mistake in writing. Now we revised as "The time series of all data were from Jan. 1951 to Dec. 2010, 720 months in total" from line129 to line130

in page6.

We sincerely hope for your satisfaction with our revision. Thank you again for your kind suggestion.

2.      lines 292, 373, 402 and 416 forecasts are evaluated up to December 2010?

Responses:Good suggestions. In previous paper, " all 4 data sets from Jan. 1951

to Jan. 2010" is mistake in writing. Now we revised as "The time series of all data were from Jan. 1951 to Dec. 2010, 720 months in total." So the lines 292, 373, 402

and 416 forecasts are surely evaluated up to December 2010.

We sincerely hope for your satisfaction with our revision. Thank you again for your kind suggestion.

3.    lines 249-253. Why does normalising the raw values avoid the overfitting problem?

Responses:Good suggestions. Now we have revised the sentences" To avoid the overfitting problem, we used $x_{nor} = \dfrac{x - x_{min}}{x_{max} - x_{min}}$ to normalize the raw value of each of the four predictors, then we used the normalized value to model and forecast." as "In order to eliminate the dimensionless relationship between variables, data standardization is to transform data from different orders of magnitude to the same order of magnitude, thus making the data comparable. So we used $x_{nor} = \dfrac{x - x_{min}}{x_{max} - x_{min}}$

to normalize the raw value of each of the four predictors, then we used the normalized value to model and forecast." from line243 to line248 in page12.

We sincerely hope for your satisfaction with our revision. Thank you again for your kind suggestion.

4.    line 254. What criterion is used to determine what are "weak items" with "small dimension coefficient".

Responses:Good suggestions. In the previous paper, we have neglected to explain the criterion is used to determine what are "weak items" with "small dimension coefficient".

In order to quantitatively compare the relative contribution of each item of our model to the evolution of the system, we calculated the relative variance contribution.

The formula is as follows: $R_i = \frac{1}{n}\sum_{j=1}^{n}[\frac{T_i^2}{\sum_{i=1}^{14}T_i^2}], i = 1, 2, ..., 14$ , Where n is the length of the data, $T_i = a_1 x_1, a_2 x_2, ..., a_{14} x_3 x_4$ is the item in the equation. According to our previous research (Hong et al., 2007), the variance contribution of the real item reflecting the performance of the model has a large proportion, while the variance contribution of the false term is almost zero, so we delete the weak items of

$R_i < 0.01$.

Now we have added the above explanation about the criterion is used to determine what are "weak items" from line250-257 in page12.

We sincerely hope for your satisfaction with our revision. Thank you again for your kind suggestion.

Responses:Good suggestions. Now we have revised the list of references carefully and make all the references complete.

We sincerely hope for your satisfaction with our revision. Thank you again for your kind suggestion.

[revised manuscript text omitted]

---

## Referee Report (RR1)

With the objective of tackling the problem of inaccurate long-term El Niño Southern Oscillation (ENSO) forecast, this study develops a new dynamical-statistical forecast model of sea surface temperature anomaly (SSTA) field. A self-memorization principle is introduced to improve the dynamic reconstruction model, making the model more appropriate for describing such chaotic systems like ENSO. Compared with six mature models published previously, the present model has an advantage in prediction precision and time length, and is a novel exploration of the ENSO forecast method. Thus, I recommend its possible publication in the OS after some minor revisions.

**Major:**

1. Line633-663: I think there is no need to spend space here to introduce the definition of overfitting and you just have to give a reference. The discussion whether the model is overfitting or not is redundant and the discussion should focus on the good forecast results of new examples. This part of the discussion can be properly streamlined.

2. Line371: Why $p$ value was in the range 5 to 15? Is this the experience of the predecessors? Or is it the result of the author's own experiment?

3. Line597-615:The purpose of this section is to show that a certain impact on the pattern of the 2nd EOF will not make our forecast unstable. But this section here occupied the space too much and I think there is no need to make a list of table5 alone. The author just says the difference among forecast results of both TC and MAPE of five different sample data are less. Because the main purpose of this paper is to prove that the prediction accuracy of SST and ENSO of the model. Therefore, it is not necessary for table5 to list separately. It is recommended that this part should be simplified.

4. Line243-249: The author introduced the standardization of the data before modelling and it is too wordy here. The author just list the standardized formula here. So the section should be suggested to be simplified.

5. Line269 and 298: The abbreviation appears for the first time, and there is no need to repeat the Pearson correlation coefficient (CC).

6. Dynamic model recommendations are consistent, the topic is dynamical-st, followed by several dynamic text in a number of tense inconsistency, it is recommended to be consistent.

**Minor:**

1.Line649: "the forecast results of the Cross-validated" should be revised as" the forecast results of the cross-validated".

2.Line 30, "dynamical reconstruction"is more appropriate

3.Line 35, "pearson" should be ''Pearson"

4.Line 36, "approximately" should be ''approximate" or "about"

5.Line 38,"but" should be "but also"

6.Line 52,"influences" should be "influence"

7.Line 57,"gradually improved" should be "gradual improvement"

8.Line 67,"large a amount of" should be " a large amount of "

9.Line 75,"advanced" should be " been advanced "

10.Line 91,"This is because" should be "It is because that"

---

## Author Response (AR2)

**Dear Editor:**

Thank you very much for providing the opportunity for us to revise our paper.

Thank you very much for your contributions to this paper. And we are all extremely grateful for having a chance to make further improvements. The major and minor suggestions of reviewer2# are very helpful for us. Reading and considering all comments of reviewer2# carefully, we have made some revisions on our paper.

Finally, we write the point-by-point response to answer the reviewer2#'s questions for better communication. If there are still any problems on the method, diction, phrasing, grammar, and spelling, please do not hesitate to tell us and we'll try our best to improve them.

Thank you again for your comments to improve our paper. Wish your journal better and better.

Yours,

Mei Hong

2018-03-18

**Responses to reviewer#2:**

All the authors are extremely grateful to you for providing your excellent comments and valuable advices for this paper. Your six major suggestions are very helpful for us. Based on your suggestions, we have made some revisions on our paper. We have streamlined the discussion and explained Why p value was in the range 5 to 15 based on your specific comments.

Thank you again for your valuable comments to improve our submission. If there are still any problems on the method, diction, phrasing, grammar, and spelling, please do not hesitate to tell us and we'll try our best to improve them.

In the following, kind comments you suggested before are in black text with corresponding actions taken by us following in blue.

Major comments:

1. Line633-663: I think there is no need to spend space here to introduce the definition of overfitting and you just have to give a reference. The discussion whether the model is overfitting or not is redundant and the discussion should focus on the good forecast results of new examples. This part of the discussion can be properly streamlined.

Responses:Good suggestions. In the previous paper, the discussion whether the model is overfitting or not is redundant and now the discussion have focused on the good forecast results of new examples. We have deleted some redundant statements to make this part of the discussion can be concise.

The specific revision can be seen from line632 to line647 in page30-31.

We sincerely hope for your satisfaction with our revision. Thank you again for your kind suggestion.

2. Line371: Why p value was in the range 5 to 15? Is this the experience of the predecessors? Or is it the result of the author's own experiment?

Responses:Good suggestions. In the reference (Cao, 1993), if the system forgets slowly, parameters and will be small and the value should be high. And he made a lot of experiments to prove that the $p$ value was in the range 5 to 15 to this system. So the conclusion of p value was in the range 5 to 15 is based on the experience of the predecessors (Cao, 1993). Now we added the explanation of why p value was in the range 5 to 15 for readers' better understanding.

The specific revision can be seen from line370 in page18.

We sincerely hope for your satisfaction with our revision. Thank you again for your kind suggestion.

3. Line597-615:The purpose of this section is to show that a certain impact on the pattern of the 2nd EOF will not make our forecast unstable. But this section here occupied the space too much and I think there is no need to make a list of table5 alone. The author just says the difference among forecast results of both TC and

MAPE of five different sample data are less. Because the main purpose of this paper is to prove that the prediction accuracy of SST and ENSO of the model. Therefore, it is not necessary for table5 to list separately. It is recommended that this part should be simplified.

Responses:Good suggestions. Based on your suggestion, we have deleted the table5 and make the discussion concise. This part had be simplified.

The specific revision can be seen from line596-609 in page29.

We sincerely hope for your satisfaction with our revision. Thank you again for your kind suggestion.

4. Line243-249: The author introduced the standardization of the data before modelling and it is too wordy here. The author just list the standardized formula here. So the section should be suggested to be simplified.

Responses:Good suggestions. Now we have just listed the standardized formula and deleted the redundant statements. So now the section has been simplified.

The specific revision can be seen from line240-246 in page11-12.

We sincerely hope for your satisfaction with our revision. Thank you again for your kind suggestion.

5. Line269 and 298: The abbreviation appears for the first time, and there is no need to repeat the Pearson correlation coefficient (CC).

Responses:Good suggestions. Now we have revised the Pearson correlation coefficient (CC) as CC and the mean absolute percentage error (MAPE) as MAPE in line294 on page14.

We sincerely hope for your satisfaction with our revision. Thank you again for your kind suggestion.

6. Dynamic model recommendations are consistent, the topic is dynamical-st, followed by several dynamic text in a number of tense inconsistency, it is recommended to be consistent.

Responses:Good suggestions. Now we have revised all the "dynamic" as the "dynamical" for the consistency.

The specific revision can be seen in the whole paper.

We sincerely hope for your satisfaction with our revision. Thank you again for your kind suggestion.

Minor comments:

1. Line649: "the forecast results of the Cross-validated" should be revised as" the forecast results of the cross-validated".

Responses:Good suggestions. Now we have revised "the forecast results of the Cross-validated" as " the forecast results of the cross-validated" in line631 in page30.

We sincerely hope for your satisfaction with our revision. Thank you again for your kind suggestion.

2. Line 30, "dynamical reconstruction" is more appropriate

Responses : Good suggestions. Now we have revised as "dynamical reconstruction" in line30 on page2.

We sincerely hope for your satisfaction with our revision. Thank you again for your kind suggestion.

3. Line 35, "pearson" should be ''Pearson"

Responses:Good suggestions. Now we have revised "pearson" as "Pearson" in line35 on page2.

We sincerely hope for your satisfaction with our revision. Thank you again for your kind suggestion.

4. Line 36, "approximately" should be ''approximate" or "about"

Responses:Good suggestions. Now we have revised "approximately" as "approximate" in line36 on page2.

We sincerely hope for your satisfaction with our revision. Thank you again for your kind suggestion.

5. Line 38,"but" should be "but also"

Responses:Good suggestions. Now we have revised "but" as "but also" in line38 on page2.

We sincerely hope for your satisfaction with our revision. Thank you again for your kind suggestion.

6. Line 52,"influences" should be "influence"

Responses:Good suggestions. Now we have revised "influences" as "influence" in line51 on page3.

We sincerely hope for your satisfaction with our revision. Thank you again for your kind suggestion.

7. Line 57,"gradually improved" should be "gradual improvement"

Responses:Good suggestions. Now we have revised "gradually improved" as "gradual improvement" in line56 on page3.

We sincerely hope for your satisfaction with our revision. Thank you again for your kind suggestion.

8. Line 67,"large a amount of" should be " a large amount of "

Responses:Good suggestions. Now we have revised "large a amount of" as "a large amount of" in line66 on page3.

We sincerely hope for your satisfaction with our revision. Thank you again for your kind suggestion.

9. Line 75,"advanced" should be " been advanced "

Responses:Good suggestions. Now we have revised"advanced" as "been advanced" in line74 on page4.

We sincerely hope for your satisfaction with our revision. Thank you again for your kind suggestion.

10. Line 91,"This is because" should be "It is because that"

Responses:Good suggestions. Now we have revised"This is because" as "It is because that" in line90 on page4.

We sincerely hope for your satisfaction with our revision. Thank you again for your kind suggestion.

The following is the a marked-up manuscript version:

[revised manuscript text omitted]

